# Unlocking Dataset Distillation with Diffusion Models

**Brian B. Moser**[1,2,4]    **Federico Raue**[2,4]    **Sebastian Palacio**[3]    **Stanislav Frolov**[1,2]

**Andreas Dengel**[1,2]

[1] University of Kaiserslautern-Landau, Kaiserslautern
[2] German Research Center for Artificial Intelligence (DFKI), Kaiserslautern
[3] ABB AG, Mannheim
[4] Equal Contribution
`first.second@dfki.de`

## Abstract

Dataset distillation seeks to condense datasets into smaller but highly representative synthetic samples. While diffusion models now lead all generative benchmarks, current distillation methods avoid them and rely instead on GANs or autoencoders, or, at best, sampling from a fixed diffusion prior. This trend arises because naive backpropagation through the long denoising chain leads to vanishing gradients, which prevents effective synthetic sample optimization. To address this limitation, we introduce Latent Dataset Distillation with Diffusion Models (LD3M), the first method to learn gradient-based distilled latents and class embeddings end-to-end through a pre-trained latent diffusion model. A linearly decaying skip connection, injected from the initial noisy state into every reverse step, preserves the gradient signal across dozens of timesteps without requiring diffusion weight fine-tuning. Across multiple ImageNet subsets at $128 \times 128$ and $256 \times 256$, LD3M improves downstream accuracy by up to 4.8 percentage points (1 IPC) and 4.2 points (10 IPC) over the prior state-of-the-art. The code for LD3M is provided at `https://github.com/Brian-Moser/prune_and_distill`.

## 1   Introduction

Large-scale datasets fuel the advancements in modern computer vision but demand substantial computational resources and raise scalability concerns [27, 2, 25, 14]. Dataset distillation emerges as a compelling solution, aiming to synthesize a small set of information-rich samples that preserve the essence of the original dataset [35, 42, 4]. While early methods operated directly in pixel space, a promising recent direction involves leveraging powerful generative models as priors [13, 1]. By optimizing compact latent codes instead of raw pixels, these approaches, exemplified by GLaD using StyleGAN-XL [5], can generate higher-resolution synthetic images ($128 \times 128$ and beyond) that generalize better across diverse network architectures.

However, GAN-based priors like in GLaD suffer from complex multi-space latent optimization and require cumbersome inversion processes for initialization [5, 43, 3, 21]. Diffusion models [18, 28], having surpassed GANs as the state-of-the-art image generators [9], represent a natural next step. Yet, *inherent* vanishing gradients severely hinder their direct application to dataset distillation. Optimizing latents *through* the long denoising chain leads to exponentially decaying gradients, which prevents effective learning of the synthetic data [19, 17].

39th Conference on Neural Information Processing Systems (NeurIPS 2025).

Mathematically, if $\mathcal{Z}$ is the initial latent code to be optimized and $\mathbf{z}_0$ the final denoised state after $T$ steps, the gradient $\partial \mathcal{L}/\partial \mathcal{Z}$ is a product of Jacobians:

$$\frac{\partial \mathcal{L}}{\partial \mathcal{Z}} = \frac{\partial \mathcal{L}}{\partial \mathbf{z}_0} \cdot \left[ \prod_{t=1}^{T} \frac{\partial \mathbf{z}_{t-1}}{\partial \mathbf{z}_t} \right] \cdot \frac{\partial \mathbf{z}_T}{\partial \mathcal{Z}}. \tag{1}$$

If the norm of each Jacobian $\partial \mathbf{z}_{t-1}/\partial \mathbf{z}_t$ is bounded by $\lambda < 1$, the product term $\prod_{t=1}^{T}(\partial \mathbf{z}_{t-1}/\partial \mathbf{z}_t)$ diminishes towards zero as $T$ increases. This gradient decay is empirically observable and significant: Our analysis confirms that in standard diffusion models, gradient norms for $\mathcal{Z}$ decrease nearly tenfold as $T$ increases from 10 to 90 (see Table 1).

Table 1: Gradient norms for the initial latent code $\mathcal{Z}$ across different maximum diffusion steps $T$ for fixed inputs. The decrease in norm as $T$ increases empirically demonstrates the vanishing gradient problem, which severely hinders the optimization of $\mathcal{Z}$ through the standard reverse diffusion process.

| $T$ | 10 | 20 | 30 | 40 | 50 | 60 | 70 | 80 | 90 |
|---|---|---|---|---|---|---|---|---|---|
| $\|\mathcal{L}/\partial \mathcal{Z}\| \times 10^4$ | 58.1 | 33.5 | 19.8 | 15.4 | 13.9 | 12.1 | 10.4 | 8.7 | 6.5 |

This critical bottleneck has forced prior diffusion-based distillation attempts to circumvent end-to-end optimization entirely, resorting instead to sampling or selecting fixed representations from pre-trained models [12, 34, 16]. These approaches, while computationally faster, forfeit the fine-grained gradient-matching optimization crucial for potential benefits like enhanced privacy of distilled data or their robustness against adversarial attacks [7, 37, 44, 10].

To unlock diffusion models for true dataset distillation, we introduce **L**atent **D**ataset **D**istillation with **D**iffusion **M**odels (LD3M). Our core contribution is a tailored modification to the reverse diffusion process (Equation 7) that introduces linearly decaying residual connections, specifically designed to enhance gradient flow for optimizing latent representations $\mathcal{Z}$ and conditioning codes $\mathbf{c}$ in the context of dataset distillation. This mechanism enables, for the first time, effective end-to-end optimization of distilled latent codes $\mathcal{Z}$ and class embeddings $\mathbf{c}$ *through* a pre-trained latent diffusion model. LD3M is readily compatible with existing distillation objectives and diffusion model architectures. Experiments across numerous ImageNet subsets demonstrate that LD3M significantly outperforms the state-of-the-art at $128 \times 128$ and $256 \times 256$ resolutions, achieving superior cross-architecture generalization, *i.e.*, 4.8 percentage points (1 IPC) and 4.2 points (10 IPC), and faster distillation times.

## 2 Preliminaries

**Dataset Distillation:** Let $\mathcal{T} = (X_r, Y_r)$, where $X_r \in \mathbb{R}^{N \times H \times W \times C}$, be a real image classification dataset and $N$ its cardinality. The goal is to compress $\mathcal{T}$ into a small synthetic set $\mathcal{S} = (X_s, Y_s)$, where $X_s \in \mathbb{R}^{M \times H \times W \times C}$, where $M$ is the total number of synthetic samples with $M = \mathcal{C} \cdot IPC$, $\mathcal{C}$ the number of classes and $IPC$ the Images Per Class (IPC). We aim to achieve $M \ll N$ with

$$\mathcal{S}^* = \arg\min_{\mathcal{S}} \mathcal{L}(\mathcal{S}, \mathcal{T}) \tag{2}$$

where $\mathcal{L}$ is a distillation loss between the distilled set $\mathcal{S}$ and the real dataset $\mathcal{T}$. Common choices for $\mathcal{L}$ include matching gradients (Dataset Condensation, DC [42]), feature distributions (Distribution Matching, DM [41]), or model parameter trajectories (Matching Training Trajectories, MTT [4]). We refer to the supplementary material for detailed definitions.

**Dataset Distillation with Generative Priors:** To improve the quality, resolution, and generalization of distilled images, recent work incorporates deep generative models as priors [5]. Instead of optimizing pixels directly, they optimize latent codes $\mathcal{Z} \in \mathbb{R}^{M \times h \times w \times d}$ with $h \cdot w \cdot d \ll H \cdot W \cdot C$, fed into a pre-trained generator $\mathcal{D}$. The optimization objective becomes

$$\mathcal{Z}^* = \arg\min_{\mathcal{Z}} \mathcal{L}(\mathcal{D}(\mathcal{Z}), \mathcal{T}). \tag{3}$$

## 3 Related Work

**GAN Priors (GLaD):** GLaD [5] pioneered distillation with generative priors using a pre-trained StyleGAN-XL [30]. While successful, it inherits GAN-specific drawbacks: (1) Optimizing the

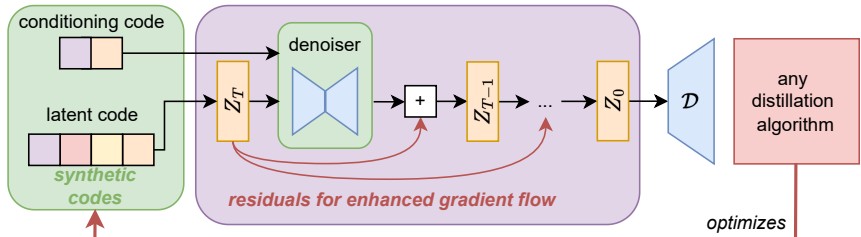

Figure 1: The LD3M Framework. Learnable latent codes $\mathcal{Z}$ and conditioning codes $\mathbf{c}$ are optimized. $\mathcal{Z}$ is noised to initialize the reverse diffusion at $\mathbf{z}_T$. A pre-trained LDM denoiser iteratively refines the state ($\mathbf{z}_t \rightarrow \mathbf{z}_{t-1}$). **Key innovation:** Residual connections (red arrows) inject $\mathbf{z}_T$ with linearly decaying weight into each step (Equation 7), enhancing gradient flow. The final latent $\mathbf{z}_0$ is decoded ($\mathcal{D}$) into images $\mathcal{S}$, which are optimized using a standard distillation algorithm.

complex, multi-level latent space ($\mathbf{W}^+$) required for high quality is computationally demanding [43]. (2) Initializing latent codes $\mathcal{Z}$ from real images $\mathbf{x}$ requires solving a costly GAN inversion problem ($\min_{\mathcal{Z}} \mathcal{L}(\mathbf{x}, \mathcal{D}(\mathcal{Z}))$) [3, 36], hindering standard initialization practices [21].

**Prior Diffusion-based Distillation Attempts:** The inherent vanishing gradient problem (§1) significantly challenges using diffusion models for end-to-end distillation. Faced with this gradient barrier, existing diffusion methods for distillation have adopted non-optimization strategies:

- **Sampling/Selection Methods:** Minimax Diffusion [16] and D4M [34] use criteria to *select* or *sample* representative latents from a diffusion model, avoiding backpropagation to latents entirely. While faster, this *bypasses gradient-based optimization that is needed for privacy or robustness*, also crucial motivations for dataset distillation [7, 37, 44, 10].
- **Autoencoder-Only Methods:** Duan et al. [12] leverage the pre-trained autoencoder from LDM but *do not utilize the diffusion process itself*. They optimize latent codes $\mathcal{Z}$ that are directly decoded by $\mathcal{D}$, essentially using only the autoencoder, not its core denoising mechanism. This simplifies optimization but fails to exploit the diffusion prior.

In contrast, LD3M is designed to directly optimize latent codes $\mathcal{Z}$ by enabling gradient flow *through* the reverse process.

**Residual Connections for Gradient Flow:** Enhancing gradient flow in deep networks is commonly addressed using residual connections, famously introduced in ResNets [17] or LSTMs [19]. Similar ideas have appeared within diffusion models; for instance, SAGE [23] used residuals connecting to the initial noise to enable adversarial latent search. While conceptually related, our contribution differs significantly: we introduce structured, *linearly decaying* residuals injected *at every step* from the *initial noisy latent* $\mathbf{z}_T$, explicitly designed for end-to-end optimization of distilled latent codes through the diffusion chain for the unique characteristics of dataset distillation.

**Decoupled Distillation Methods:** Distinct from methods optimizing synthetic data via generative priors, another line of work decouples distillation from end-to-end training. Methods like SRe2L [40] and others [39, 32] leverage statistics (e.g., from BatchNorm) of pre-trained networks to recover informative images, offering scalability benefits but following a fundamentally different optimization strategy than gradient-matching approaches like LD3M.

# 4   Latent Dataset Distillation with Diffusion Models (LD3M)

Our approach enables end-to-end optimization of synthetic data directly through a pre-trained LDM. As shown in Figure 1, LD3M optimizes both initial latent codes $\mathcal{Z}$ and conditioning codes $\mathbf{c}$. These are processed by a modified reverse diffusion process to generate expressive latent states $\mathbf{z}_0$, which are then decoded into images, $\mathcal{S} = \mathcal{D}(\mathbf{z}_0)$. This section details the core components: our modified diffusion sampling process designed to boost gradient flow (§4.1), the efficient initialization strategy (§4.2), and the gradient checkpointing used for memory efficiency (§4.3).

## 4.1 Sampling Process

We leverage a pre-trained LDM [28] without fine-tuning its weights. Standard LDM operation involves a reverse diffusion process $p_\theta$ that iteratively denoises a state $\mathbf{z}_t$, starting from noise $\mathbf{z}_T$, conditioned on an embedding $\mathbf{c}$ (typically derived from class labels) [38, 26]. Each step $t$ predicts a less noisy state $\mathbf{z}_{t-1}$ based on the current state $\mathbf{z}_t$ and condition $\mathbf{c}$. The standard update rule [18, 29] calculates the subsequent state $\mathbf{z}_{t-1}$ using the predicted mean $\mu_\theta$ and variance $\sigma_t^2$:

$$\mu_\theta(\mathbf{c}, \mathbf{z}_t, \gamma_t) = \frac{1}{\sqrt{\alpha_t}} \left( \mathbf{z}_t - \frac{1 - \alpha_t}{\sqrt{1 - \gamma_t}} f_\theta(\mathbf{c}, \mathbf{z}_t, \gamma_t) \right) \tag{4}$$

$$\mathbf{z}_{t-1} \leftarrow \mu_\theta(\mathbf{c}, \mathbf{z}_t, \gamma_t) + \sigma_t^2 \varepsilon_t, \tag{5}$$

where $f_\theta$ is the LDM's pre-trained noise prediction network (usually a U-Net), $\alpha_t$ and $\gamma_t$ relate to the noise schedule, and $\varepsilon_t \sim \mathcal{N}(\mathbf{0}, \mathbf{I})$ is random noise added at step $t$.

---

**Algorithm 1** Latent Dataset Distillation with Diffusion Models (LD3M)

---

**Input:** randomly selected collection $X_s$, pre-trained encoder $\mathcal{E}$, pre-trained decoder $\mathcal{D}$, pre-trained denoiser $\mu_\theta$ with frozen parameters $\theta$, noise levels $\sigma_t$.

> $\mathcal{Z} = \mathcal{E}(X_s)$
> $\mathbf{z}_T \sim q(\mathbf{z}_T \mid \mathcal{Z})$
> **for** $t = T, \ldots, 1$ **do**
> $\quad \varepsilon_t \sim \mathcal{N}(\mathbf{0}, \mathbf{I})$
> $\quad \mathbf{z}_{t-1} \leftarrow \left( (1 - \frac{t}{T}) \cdot \mu_\theta(\mathbf{c}, \mathbf{z}_t, \gamma_t) + \frac{t}{T} \cdot \mathbf{z}_T \right) + \sigma_t^2 \varepsilon_t$
> **end for**
> $X_{\text{syn}} \leftarrow \mathcal{D}(\mathbf{z}_0)$
> **Return:** $X_{\text{syn}}$

---

For dataset distillation, our goal is to learn the optimal initial latent representations $\mathcal{Z}$ and conditioning codes $\mathbf{c}$ that minimize the distillation loss $\mathcal{L}$ between the synthetic images and the target dataset:

$$\mathcal{Z}^*, \mathbf{c}^* = \underset{\mathcal{Z}, \mathbf{c}}{\arg\min} \, \mathcal{L}(\mathcal{D}[p_\theta(\mathbf{z}_0|\mathbf{z}_T, \mathbf{c})], \mathcal{T}), \quad \text{where } \mathbf{z}_T \sim q(\mathbf{z}_T|\mathcal{Z}). \tag{6}$$

Here, $p_\theta(\mathbf{z}_0|\mathbf{z}_T, \mathbf{c})$ denotes the final state $\mathbf{z}_0$ resulting from the $T$-step reverse process starting from $\mathbf{z}_T$, which itself is a noised version of the learnable $\mathcal{Z}$ obtained via the forward process $q$. We want to highlight that the sampling of $\varepsilon$ introduces stochasticity during inference. We fix, however, for one sampling phase, the residual variable $\mathbf{z}_T$ to be constantly the same.

To effectively minimize distillation losses like gradient or trajectory matching (Equation 6), which rely on fine-grained alignment between synthetic and real data processing, requires guiding the generative process. While conditioning $\mathbf{c}$ provides class guidance, optimizing the initial latent $\mathcal{Z}$ offers the necessary degrees of freedom to shape the generated sample $\mathcal{D}(\mathbf{z}_0)$ precisely. Relying solely on fixed $\mathcal{Z}$ or only optimizing $\mathbf{c}$ proved insufficient empirically, yielding suboptimal results resembling simple LAION-5B [31] data retrieval similar to D4M [34].

Vanishing gradients inherent in backpropagating through the $T$ steps of Equation 5 present the primary obstacle to optimizing Equation 6 [19]. To overcome this, we introduce a simple yet effective modification to the reverse step, injecting a residual connection from the initial state $\mathbf{z}_T$:

$$\mathbf{z}_{t-1} \leftarrow \underbrace{\left( (1 - \frac{t}{T}) \cdot \mu_\theta(\mathbf{c}, \mathbf{z}_t, \gamma_t) + \frac{t}{T} \cdot \mathbf{z}_T \right)}_{\text{Modified Mean}} + \sigma_t^2 \varepsilon_t. \tag{7}$$

Crucially, while this modification deviates from standard sampling aimed at matching the distribution of real images, its purpose here is to enable gradient flow for distillation optimization, a task where downstream utility, not photorealism [42, 4], is paramount (see supplementary material for detailed discussion). In other words, our method breaks the sampler's fidelity to the original data distribution in order to better achieve the distillation objective of matching feature distributions.

In conclusion, the modification replaces the standard predicted mean $\mu_\theta$ with a weighted average of $\mu_\theta$ and the initial state $\mathbf{z}_T$. The weight of $\mathbf{z}_T$ decreases linearly from 1 (at $t = T$) to nearly 0 (as $t \to 0$). This creates a direct pathway for gradients from the loss $\mathcal{L}$ (computed using $\mathbf{z}_0$) back to $\mathbf{z}_T$,

and thus to the learnable $\mathcal{Z}$, bypassing the long chain of Jacobian products that causes gradients to vanish. The enhanced gradient flow to $\mathcal{Z}$ can be conceptually represented as:

$$\frac{\partial \mathcal{L}}{\partial \mathcal{Z}} = \sum_{t=1}^{T} \underbrace{\left(1 - \frac{T-1}{T}\right) \cdot \left[\frac{\partial \mathcal{L}}{\partial \mathbf{z}_t} \cdot \frac{\partial \mathbf{z}_t}{\partial \mathbf{z}_{t-1}} \cdot \dots \cdot \frac{\partial \mathbf{z}_0}{\partial \mathcal{Z}}\right]}_{\text{Original (Decaying) Path}} + \underbrace{\left(\tfrac{t}{T}\right) \cdot \left[\frac{\partial \mathcal{L}}{\partial \mathbf{z}_{t-1}} \frac{\partial \mathbf{z}_{t-1}}{\partial \mathbf{z}_T} \frac{\partial \mathbf{z}_T}{\partial \mathcal{Z}}\right]}_{\text{Enhanced Path via Skip Connection}}. \quad (8)$$

A comprehensive description can be found in Algorithm 1.

**Notes on Markovian Property:** $\mathbf{z}_{t-1}$ depends on $\mathbf{z}_t$ and $\mathbf{z}_T$, but not on any earlier states such as $\mathbf{z}_{t+1}$, $\mathbf{z}_{t+2}$, and so on. Therefore, the probability distribution for $\mathbf{z}_{t-1}$ only depends on $\mathbf{z}_t$ and the fixed initial state $\mathbf{z}_T$, which is constant throughout the diffusion process. Thus, we have: $p(\mathbf{z}_{t-1}|\mathbf{z}_t, \mathbf{z}_{t+1}, \dots, \mathbf{z}_T) = p(\mathbf{z}_{t-1}|\mathbf{z}_t, \mathbf{z}_T)$. This confirms that LD3M remains Markovian.

**Notes on Generalisability:** Our gradient enhancement technique (Equation 7) applies to various diffusion model architectures beyond LDMs, suggesting broader potential for future work. In this study, we focused on LDMs primarily as a proof of concept, leveraging readily available pre-trained models and LDM's foundational role in latent-space diffusion.

## 4.2 Efficient Latent Code Initialization

Standard practice in dataset distillation initializes synthetic data using real images from the target classes [21]. GAN-based methods like GLaD face a significant challenge here, requiring complex and costly GAN inversion techniques to find latent codes $\mathcal{Z}$ that reconstruct target real images $\mathbf{x}$ [36].

LD3M benefits immensely from the autoencoder structure inherent in LDMs. We initialize the learnable latent codes $\mathcal{Z}_{\text{init}}$ simply by encoding a small, randomly selected set of real images $X_s$ using the pre-trained image encoder. Similarly, the initial conditioning codes $\mathbf{c}_{\text{init}}$ are obtained using the pre-trained class embedding network.

## 4.3 Memory Efficiency via Gradient Checkpointing

Optimizing through the $T$ steps of the reverse diffusion process, even with our modification, can be memory-intensive. Following GLaD [5], we employ gradient checkpointing [6] to manage VRAM usage. The procedure involves:

1. Perform the forward pass $\mathcal{S} = \mathcal{D}(p_\theta(\mathbf{z}_0|\mathbf{z}_T, \mathbf{c}))$ *without* storing intermediate activations.
2. Calculate the distillation loss $\mathcal{L}(\mathcal{S}, \mathcal{T})$ and the gradient with respect to the output, $\partial \mathcal{L}/\partial \mathcal{S}$.
3. To compute the gradient $\partial \mathcal{L}/\partial \mathcal{Z}$ (and $\partial \mathcal{L}/\partial \mathbf{c}$), recompute the necessary segments of the forward pass through the diffusion process and decoder $\mathcal{D}$, storing only the activations needed for the immediate backward pass segment.

This avoids storing the full computation graph, which GLaD also exploits for a single generator pass.

# 5 Experiments

We evaluate LD3M against relevant baselines, primarily the state-of-the-art latent generative prior method GLaD [5], following its experimental setup for fair comparison. We conduct extensive experiments on 10 diverse 10-class subsets of ImageNet-1k [8] at $128 \times 128$ (IPC=1, IPC=10) and $256 \times 256$ (IPC=1) resolutions, as well as CIFAR-10. Key implementation details are in §5.1; full hyperparameters and setup details are in the supplementary material.

## 5.1 Setup Details

**Datasets & Evaluation:** We use ImageNet subsets (ImNet-A to E, ImNette, ImWoof, Birds, Fruits, Cats) from [5, 20, 4] and CIFAR-10. Following standard protocol, we distill datasets using DC [42], DM [41], or MTT [4] and evaluate by training unseen architectures (AlexNet [22], VGG-11 [33], ResNet-18 [17], ViT [11]) from scratch on the distilled set, reporting mean test accuracy ($\pm$ std. dev.) over 5 runs.

Table 2: CIFAR-10 comparisons with GLaD on IPC=1 and SRe2L/D4M on IPC=50.

| | (a) IPC=1 | | | | | | | (b) IPC=50 | |
|---|---|---|---|---|---|---|---|---|---|
| Dist. | Method | AlexNet | ResNet18 | VGG11 | ViT | Average | | Method | ConvNet |
| DC | pixel space | $25.9_{\pm0.2}$ | $27.3_{\pm0.5}$ | $28.0_{\pm0.5}$ | $22.9_{\pm0.3}$ | $26.0_{\pm0.4}$ | | SRe2L [40] | $60.2^{\dagger}$ |
| | GLaD (rand G) | $\mathbf{30.1}_{\pm0.5}$ | $27.3_{\pm0.2}$ | $28.0_{\pm0.9}$ | $21.2_{\pm0.6}$ | $26.6_{\pm0.5}$ | | D4M [34] | $72.8^{*}$ |
| | GLaD (trained G) | $26.0_{\pm0.7}$ | $\mathbf{27.6}_{\pm0.6}$ | $28.2_{\pm0.6}$ | $23.4_{\pm0.2}$ | $26.3_{\pm0.5}$ | | **LD3M (Ours)** | **73.2** |
| | **LD3M** (trained G) | $27.2_{\pm0.8}$ | $26.6_{\pm0.9}$ | $\mathbf{31.5}_{\pm0.3}$ | $\mathbf{29.0}_{\pm0.2}$ | $\mathbf{28.6}_{\pm0.6}$ | | | |
| DM | pixel space | $22.9_{\pm0.2}$ | $22.2_{\pm0.7}$ | $23.8_{\pm0.5}$ | $21.3_{\pm0.5}$ | $22.6_{\pm0.5}$ | | | |
| | GLaD (rand G) | $23.7_{\pm0.3}$ | $21.7_{\pm1.0}$ | $24.3_{\pm0.8}$ | $21.4_{\pm0.2}$ | $22.8_{\pm0.6}$ | | | |
| | GLaD (trained G) | $25.1_{\pm0.5}$ | $22.5_{\pm0.7}$ | $24.8_{\pm0.8}$ | $23.0_{\pm0.1}$ | $23.8_{\pm0.5}$ | | | |
| | **LD3M** (trained G) | $\mathbf{27.2}_{\pm0.4}$ | $\mathbf{23.0}_{\pm0.7}$ | $\mathbf{25.4}_{\pm0.4}$ | $\mathbf{23.8}_{\pm0.3}$ | $\mathbf{24.9}_{\pm0.5}$ | | | |

$\dagger$ Reported [40] at IPC=1K; Decoupled method.
$*$ Reported [34]; Sampling-based Diffusion.

Table 3: Cross-architecture performance (%) with 1 IPC on ImageNet subsets ($128 \times 128$). LD3M (**bold**) generally achieves the best results within each algorithm and overall best (blue) compared to Pixel Space and GLaD. For instance, LD3M improves DC, MTT, and DM by +3.76%, +5.68%, and +16.34%, respectively.

| Distil. Space | Alg. | All | ImNet-A | ImNet-B | ImNet-C | ImNet-D | ImNet-E | ImNette | ImWoof | ImNet-Birds | ImNet-Fruits | ImNet-Cats |
|---|---|---|---|---|---|---|---|---|---|---|---|---|
| pixel space | MTT | $25.4_{\pm2.3}$ | $33.4_{\pm1.5}$ | $34.0_{\pm3.4}$ | $31.4_{\pm3.4}$ | $27.7_{\pm2.7}$ | $24.9_{\pm1.8}$ | $24.1_{\pm1.8}$ | $16.0_{\pm1.2}$ | $25.5_{\pm3.0}$ | $18.3_{\pm2.3}$ | $18.7_{\pm1.5}$ |
| | DC | $27.9_{\pm1.6}$ | $38.7_{\pm4.2}$ | $38.7_{\pm1.0}$ | $33.3_{\pm1.9}$ | $26.4_{\pm1.1}$ | $27.4_{\pm0.9}$ | $28.2_{\pm1.4}$ | $17.4_{\pm1.2}$ | $28.5_{\pm1.4}$ | $20.4_{\pm1.5}$ | $19.8_{\pm0.9}$ |
| | DM | $19.2_{\pm1.0}$ | $27.2_{\pm1.2}$ | $24.4_{\pm1.1}$ | $23.0_{\pm1.4}$ | $18.4_{\pm0.7}$ | $17.7_{\pm0.9}$ | $20.6_{\pm0.7}$ | $14.5_{\pm0.9}$ | $17.8_{\pm0.8}$ | $14.5_{\pm1.1}$ | $14.0_{\pm1.1}$ |
| GLaD | MTT | $29.0_{\pm1.2}$ | $39.9_{\pm1.2}$ | $39.4_{\pm1.3}$ | $\mathbf{34.9}_{\pm1.1}$ | $30.4_{\pm1.5}$ | $29.0_{\pm1.1}$ | $30.4_{\pm1.5}$ | $17.1_{\pm1.1}$ | $28.2_{\pm1.1}$ | $21.1_{\pm1.2}$ | $19.6_{\pm1.2}$ |
| | DC | $29.8_{\pm1.3}$ | $41.8_{\pm1.7}$ | $\mathbf{42.1}_{\pm1.2}$ | $35.8_{\pm1.4}$ | $28.0_{\pm0.8}$ | $29.3_{\pm1.3}$ | $31.0_{\pm1.6}$ | $17.8_{\pm1.1}$ | $29.1_{\pm1.0}$ | $22.3_{\pm1.6}$ | $21.2_{\pm1.4}$ |
| | DM | $22.4_{\pm1.4}$ | $31.6_{\pm1.4}$ | $31.3_{\pm3.9}$ | $26.9_{\pm1.2}$ | $21.5_{\pm1.0}$ | $20.4_{\pm0.8}$ | $21.9_{\pm1.1}$ | $15.2_{\pm0.9}$ | $18.2_{\pm1.0}$ | $20.4_{\pm1.6}$ | $16.1_{\pm0.7}$ |
| **LD3M** | MTT | $\mathbf{30.4}_{\pm1.3}$ | $\mathbf{40.9}_{\pm1.1}$ | $\mathbf{41.6}_{\pm1.7}$ | $34.1_{\pm1.7}$ | $\mathbf{31.5}_{\pm1.2}$ | $\mathbf{30.1}_{\pm1.3}$ | $\mathbf{32.0}_{\pm1.3}$ | $\mathbf{19.9}_{\pm1.2}$ | $\mathbf{30.4}_{\pm1.5}$ | $21.4_{\pm1.1}$ | $\mathbf{22.1}_{\pm1.0}$ |
| | DC | $\mathbf{30.9}_{\pm1.2}$ | $\mathbf{42.3}_{\pm1.3}$ | $42.0_{\pm1.1}$ | $\mathbf{37.1}_{\pm1.8}$ | $29.7_{\pm1.3}$ | $\mathbf{31.4}_{\pm1.1}$ | $\mathbf{32.9}_{\pm1.2}$ | $18.9_{\pm0.6}$ | $30.2_{\pm1.4}$ | $22.6_{\pm1.3}$ | $21.7_{\pm0.8}$ |
| | DM | $25.9_{\pm1.2}$ | $35.8_{\pm1.1}$ | $34.1_{\pm1.0}$ | $30.3_{\pm1.2}$ | $24.7_{\pm1.0}$ | $24.5_{\pm0.9}$ | $26.8_{\pm1.7}$ | $18.1_{\pm0.7}$ | $23.0_{\pm1.8}$ | $\mathbf{24.5}_{\pm1.9}$ | $17.0_{\pm1.1}$ |

Table 4: Cross-architecture performance (%) with 10 IPC on ImageNet A-E ($128 \times 128$). LD3M (**bold**, blue) consistently outperforms Pixel Space and GLaD with, for instance, an improvement of +2.52% and +3.46% with DC and DM, respectively.

| Distil. Space | Alg. | All | ImNet-A | ImNet-B | ImNet-C | ImNet-D | ImNet-E |
|---|---|---|---|---|---|---|---|
| pixel space | DC | $42.3_{\pm3.5}$ | $52.3_{\pm0.7}$ | $45.1_{\pm8.3}$ | $40.1_{\pm7.6}$ | $36.1_{\pm0.4}$ | $38.1_{\pm0.4}$ |
| | DM | $44.4_{\pm0.5}$ | $52.6_{\pm0.4}$ | $50.6_{\pm0.5}$ | $47.5_{\pm0.7}$ | $35.4_{\pm0.4}$ | $36.0_{\pm0.5}$ |
| GLaD | DC | $45.9_{\pm1.0}$ | $53.1_{\pm1.4}$ | $50.1_{\pm0.6}$ | $48.9_{\pm1.1}$ | $38.9_{\pm1.0}$ | $38.4_{\pm0.7}$ |
| | DM | $45.8_{\pm0.6}$ | $52.8_{\pm1.0}$ | $51.3_{\pm0.6}$ | $\mathbf{49.7}_{\pm0.4}$ | $36.4_{\pm0.4}$ | $38.6_{\pm0.7}$ |
| **LD3M** | DC | $\mathbf{47.1}_{\pm1.2}$ | $\mathbf{55.2}_{\pm1.0}$ | $\mathbf{51.8}_{\pm1.4}$ | $\mathbf{49.9}_{\pm1.3}$ | $\mathbf{39.5}_{\pm1.0}$ | $\mathbf{39.0}_{\pm1.3}$ |
| | DM | $\mathbf{47.3}_{\pm2.1}$ | $\mathbf{57.0}_{\pm1.3}$ | $\mathbf{52.3}_{\pm1.1}$ | $48.2_{\pm4.9}$ | $\mathbf{39.5}_{\pm1.5}$ | $\mathbf{39.4}_{\pm1.8}$ |

**Models:** We use ConvNet-5/ConvNet-6 [15] for distillation. As in GLaD [5], we use AlexNet [22], VGG-11 [33], ResNet-18 [17], and a Vision Transformer [11] for evaluating the distilled dataset quality for unseen architectures. For LD3M, we use the public ImageNet pre-trained LDM [28] with its $2\times$ compression autoencoder, without fine-tuning. Default diffusion steps $T = 10$ ($128^2$) or $T = 20$ ($256^2$) are used unless specified. For GLaD comparison, the ImageNet pre-trained StyleGAN-XL [30] is used.

## 5.2 Results

**Main Results: LD3M Outperforms State-of-the-Art.** We first establish LD3M's effectiveness against the primary baseline GLaD and other state-of-the-art methods on CIFAR-10 (Table 2). At IPC=1, LD3M matches or exceeds GLaD using DC/DM distillation. The sampling-based D4M [34] delivered $\approx$10% accuracy for all tested models. At IPC=50, LD3M surpasses reported results from D4M and decoupled SRe2L [40] with MTT (though optimization and IPC differ, see caption). We subsequently focus our main analysis on GLaD [5] as the leading state-of-the-art baseline for latent generative prior gradient-matching distillation.

Moving to the more challenging ImageNet subsets, LD3M consistently outperforms GLaD across resolutions and IPC values. For IPC=1 at $128 \times 128$ (Table 3), LD3M achieves the best overall

Table 5: Cross-architecture performance (%) for $256 \times 256$ images (DC, IPC=1) using different generator initializations (ImageNet, FFHQ, Random). LD3M outperforms GLaD across initializations. Best results marked **bold**, top 3 blue. For example, LD3M improves by roughly +2 p.p. on average, whereas it improves the performance by roughly +7 p.p. compared to pixel space.

| Distil. Space | All | ImNet-A | ImNet-B | ImNet-C | ImNet-D | ImNet-E |
|---|---|---|---|---|---|---|
| pixel space | 29.5±3.1 | 38.3±4.7 | 32.8±4.1 | 27.6±3.3 | 25.5±1.2 | 23.5±2.4 |
| GLaD (ImageNet) | 34.4±2.6 | 37.4±5.5 | 41.5±1.2 | 35.7±4.0 | 27.9±1.0 | 29.3±1.2 |
| GLaD (Random) | 34.5±1.6 | 39.3±2.0 | 40.3±1.7 | 35.0±1.7 | 27.9±1.4 | 29.8±1.4 |
| GLaD (FFHQ) | 34.0±2.1 | 38.3±5.2 | 40.2±1.1 | 34.9±1.1 | 27.2±0.9 | 29.4±2.1 |
| **LD3M** (ImageNet) | 36.3±1.6 | **42.1±2.2** | **42.1±1.5** | 35.7±1.7 | 30.5±1.4 | 30.9±1.2 |
| **LD3M** (Random) | **36.5±1.6** | 42.0±2.0 | 41.9±1.7 | **37.1±1.4** | 30.5±1.5 | **31.1±1.4** |
| **LD3M** (FFHQ) | 36.3±1.5 | 42.0±1.6 | 41.9±1.6 | 36.5±2.2 | **30.5±0.9** | 30.6±1.1 |

Table 6: Ablation: Impact of Diffusion (DC, IPC=1). Comparing LD3M (w/ diffusion) against GLaD and LD3M using only the autoencoder (w/o diffusion). Full LD3M consistently outperforms both.

| Method | All | ImNet-A | ImNet-B | ImNet-C | ImNet-D | ImNet-E |
|---|---|---|---|---|---|---|
| GLaD | 35.4±1.3 | 41.8±1.7 | **42.1±1.2** | 35.8±1.4 | 28.0±0.8 | 29.3±1.3 |
| LD3M (w/o diffusion) | 35.3±1.3 | 40.6±1.9 | 41.9±1.1 | 35.3±1.0 | 29.4±1.4 | 29.5±1.3 |
| LD3M (w/ diffusion) | **36.5**±1.3 | **42.3**±1.3 | 42.0±1.1 | **37.1**±1.8 | **29.7**±1.3 | **31.4**±1.1 |
| | +1.2+0.0 | +1.7-0.6 | +0.1+0.0 | +1.8+0.8 | +0.3-0.1 | +1.9-0.2 |

average accuracy in 9/10 subsets, significantly boosting performance particularly for DM (+16.3% avg.). This advantage persists at IPC=10 (Table 4), where LD3M improves average accuracy over GLaD by +2.5% (DC) and +3.5% (DM), reaching over 47% overall. The performance gains extend to $256 \times 256$ resolution (Table 5), where LD3M surpasses GLaD by +6% on average, even when comparing differently pre-trained generators (ImageNet, FFHQ, random), highlighting the robustness of leveraging the LDM prior via LD3M.

**Ablation Studies: Why LD3M Works.** The diffusion process itself is crucial for LD3M's improved performance over autoencoder-only approaches. Table 6 provides empirical evidence: removing the diffusion steps ("w/o diffusion", akin to *Duan et al.* [12]) results in performance comparable to GLaD (35.3% avg.), which is significantly lower than full LD3M utilizing the diffusion process (36.5% avg.). This +1.2 percentage points highlight that simply using the LDM's autoencoder is insufficient; successfully optimizing *through* the reverse process is necessary to unlock the performance benefits observed, validating our core approach over simpler AE-only methods. Naturally, we can expect similar limitations with few- or one-step diffusion models *in the context of dataset distillation*.

Furthermore, both learnable latents and our proposed gradient enhancement drive LD3M's effectiveness. Table 7 dissects these contributions: optimizing only conditioning **c** yields poor results (15.8% avg.), but making the latent $\mathcal{Z}$ learnable provides a substantial boost (22.3% avg.), confirming the necessity of latent optimization motivated in §4.1. Critically, however, this alone does not surpass GLaD; only by subsequently adding our enhanced gradient flow (Equation 7) does LD3M achieve its SOTA performance (28.1% avg., vs. 26.6% for GLaD). This clearly demonstrates that while learnable latents offer vital degrees of freedom, they cannot be effectively optimized through standard diffusion backpropagation; the enhanced gradient flow enabled by our residual connection (Equation 7) is the key component that makes end-to-end optimization through diffusion truly effective for this task.

**Robustness and Practical Considerations.** LD3M exhibits robustness in various aspects. For instance, standard initialization using real images encoded via the LDM's encoder significantly benefits DC and DM performance compared to Gaussian noise initialization (Table 8), while being vastly simpler than GLaD's GAN inversion (§4.2). Visually (Figure 2 and Figure 3), LD3M generates abstract but class-informative images that appear more consistent across different LDM pre-training datasets (ImageNet, FFHQ, Random) compared to GLaD. Our analysis of diffusion steps $T$ (Figure 4) reveals an optimal performance/runtime trade-off around $T = 10 - 40$.

Table 7: Ablation: Different LDM variations (MTT on ImageNette and IPC=1). Tested was the distillation by making only the conditioning learnable (**c**), one with also learnable latent representation ($\mathcal{Z}$), and lastly, one which incorporates our modified reverse process (Equation 7).

| Method | All | AlexNet | VGG-11 | ResNet-18 | ViT |
|---|---|---|---|---|---|
| GLaD | 26.6±1.6 | 28.7±0.3 | **29.2±1.2** | **30.8±2.9** | 17.8±1.5 |
| LDM learnable conditioning (**c**) | 15.8±1.5 | 14.2±2.6 | 15.1 ±1.6 | 16.5±4.9 | 16.8±4.0 |
| + learnable latent code ($\mathcal{Z}$) | 22.3±2.0 | 22.8±2.0 | 26.3±0.9 | 23.4±3.2 | 17.5±2.0 |
| + enhanced gradient flow (Eq. 7) | **28.1±3.3** | **29.2±1.9** | **29.2±1.2** | 30.6±1.3 | **25.1±1.7** |

Table 8: Ablation: Impact of Initialization (ImageNette, IPC=1, 5K iter.). Compares initializing $\mathcal{Z}$ and **c** from Gaussian noise vs. encoded real images across different distillation algorithms.

| Dist. Method | Dist. Space | All | AlexNet | ResNet18 | VGG11 | ViT |
|---|---|---|---|---|---|---|
| MTT | Gauss. noise | 31.0±1.4 | 28.7±1.6 | 34.1±1.5 | 32.2±0.6 | 29.1±1.8 |
| | random image | **32.0±1.3** | **30.1±1.4** | **35.6±1.4** | **32.2±0.3** | **30.0±1.2** |
| DC | Gauss. noise | 13.1±2.1 | 13.1±1.5 | 11.6±1.8 | 13.8±2.2 | 13.7±2.7 |
| | random image | **32.9±2.1** | **31.6±1.3** | **30.4±0.6** | **31.8±1.2** | **37.7±1.5** |
| DM | Gauss. noise | 13.4±1.8 | 13.4±2.0 | 12.4±1.8 | 13.4±1.4 | 14.4±2.1 |
| | random image | **26.8±1.7** | **31.9±1.3** | **23.2±2.2** | **25.9±2.0** | **26.1±1.4** |

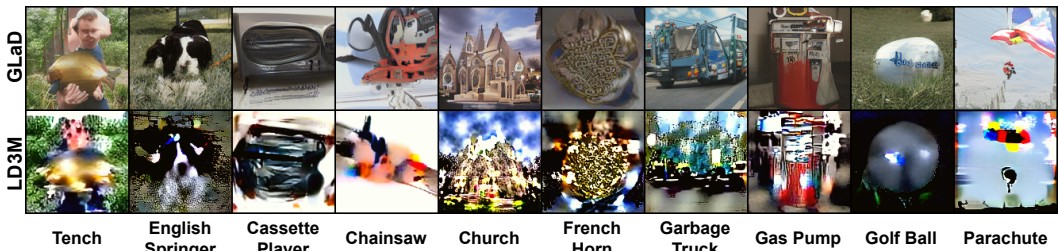

Figure 2: Visual comparison of LD3M versus GLaD (MTT, ImageNette, 1K iter.). GLaD outputs tend toward smooth, photo-realistic textures but can blur class-defining details, whereas LD3M produces bolder, higher-contrast shapes that highlight key discriminative features (*e.g.*, wing contours, beak outline). This abstraction trade-off suggests LD3M prioritizes core class signals over pixel-perfect fidelity, which empirically enhances downstream model generalization, contrasting claims made by sampling-based methods like D4M [34].

LD3M also offers computational flexibility. Our analysis indicates an optimal balance between performance and distillation time occurs with around $T = 10 - 40$ diffusion steps (Figure 4). Using $T = 20$, LD3M requires slightly less peak GPU memory on an A100-40GB compared to GLaD (29.4GB vs 31.2GB) and completes the distillation process faster (574 min vs 693 min). This computational efficiency, coupled with the inherent flexibility to adjust the number of diffusion steps ($T$) based on available resources, enhances LD3M's practical appeal.

## 6 Conclusion

We addressed the critical challenge preventing end-to-end dataset distillation through powerful diffusion models: vanishing gradients across the long denoising chain. We introduced **LD3M**, which unlocks optimization through diffusion priors via a simple yet effective modification: injecting linearly decaying residual connections from the initial noisy state into each reverse step. This novel approach enhances gradient flow sufficiently to effectively learn both latent codes ($\mathcal{Z}$) and conditioning (**c**) without altering pre-trained model weights. This direct optimization contrasts sharply with previous diffusion distillation methods that circumvented the gradient challenge by relying solely on sampling fixed representations [34, 16].

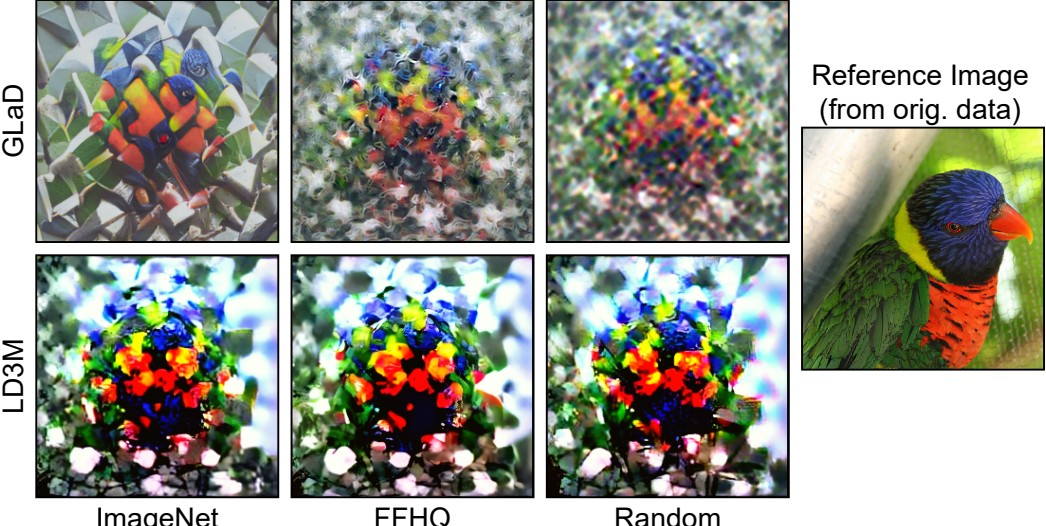

Figure 3: Example $256 \times 256$ images of a distilled class (ImageNet-B: Lorikeet) with differently initialized generators GLaD and LD3M. The various initializations, *i.e.*, which dataset was used for training the generators, are denoted at the bottom. We used DC as distillation algorithm.

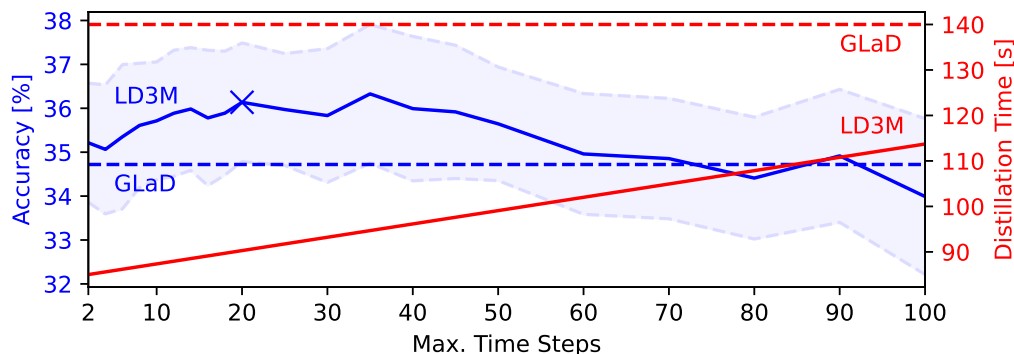

Figure 4: Accuracy vs. Distillation Time Trade-off with Diffusion Steps $T$ (ImageNet A-E avg., MTT, IPC=1). LD3M performance (blue line, mean $\pm$ std) peaks around $T = 35$. GLaD baseline (dashed lines) and optimal trade-off (X) shown for reference.

Our experiments (§5) demonstrate LD3M's clear advantages. It significantly outperforms the state-of-the-art GAN-prior method, GLaD, on diverse ImageNet subsets at $128 \times 128$ and $256 \times 256$ resolutions, improving cross-architecture generalization by up to 4.8 percentage points (IPC=1) and 4.2 points (IPC=10). Ablation studies confirmed that both leveraging the diffusion process and our specific gradient enhancement are crucial for this success. Furthermore, LD3M offers practical benefits: vastly simpler initialization than GAN inversion (§4.2) and faster overall distillation times compared to GLaD (§5).

## 7   Future Work

By successfully enabling gradient-based optimization through diffusion models, LD3M paves the way for leveraging state-of-the-art generative diffusion models for creating highly effective and compact distilled datasets. Future work should explore alternative residual formulations and integration with fast samplers like DPM-Solver [24]. Scaling and evaluation on larger benchmarks like ImageNet-1K also remain important next steps, contingent on computational resources.

## Acknowledgments

This work was supported by the EU project SustainML (Grant 101070408) and by the BMBF Project Albatross (Grant 01IW24002). All compute was done thanks to the Pegasus cluster at DFKI.

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

## Supplementary Material

## A   Definitions of DC, DM, and MTT

**Dataset Condensation (DC)** ensures alignment by deriving the gradients via a classification error [42]. It calculates the loss on real ($\ell^{\mathcal{T}}$) and the respective synthetic data ($\ell^{\mathcal{S}}$). Next, it minimizes the distance between the gradients of both network instances. More concretely,

$$\mathcal{L}_{DC} = 1 - \frac{\nabla_\theta \ell^{\mathcal{S}}(\theta) \cdot \nabla_\theta \ell^{\mathcal{T}}(\theta)}{\|\nabla_\theta \ell^{\mathcal{S}}(\theta)\| \, \|\nabla_\theta \ell^{\mathcal{T}}(\theta)\|}. \tag{9}$$

**Distribution Matching (DM)** obtains gradients by minimizing the logits on the real and synthetic datasets. It enforces the feature extractor (ConvNet) to produce similar features for real and synthetic images [41]. The distribution matching loss is

$$\mathcal{L}_{DM} = \sum_c \left\| \frac{1}{|\mathcal{T}_c|} \sum_{\mathbf{x} \in \mathcal{T}_c} \psi(\mathbf{x}) - \frac{1}{|\mathcal{S}_c|} \sum_{\mathbf{s} \in \mathcal{S}_c} \psi(\mathbf{s}) \right\|^2, \tag{10}$$

where $\mathcal{T}_c, \mathcal{S}_c$ are the real and synthetic images for a class $c$.

**Matching Training Trajectories (MTT)** concentrates on the trajectory of network parameters [4]. In more detail, MTT exploits several trained instances of a model, called experts, and stores the training trajectory of parameters $\{\theta_t^*\}_0^T$ at predetermined intervals, called expert trajectories. For dataset distillation, MTT samples a random set of parameters $\theta_t^*$ from the trajectory at a given timestamp. Next, it trains a new network, $\hat{\theta}_{t+N}$, initialized with the parameters on the respective synthetic images (for $N$ iterations). Finally, the distance between the trajectory on the real dataset, $\theta_{t+M}^*$ with $M$ steps, and the trajectory on the synthetic one, $\hat{\theta}_{t+N}$, is minimized. As a result, MTT tries to mimic the original dataset's training path (trajectory of parameters) with the synthetic images:

$$\mathcal{L}_{MTT} = \frac{\|\hat{\theta}_{t+N} - \theta_{t+M}^*\|^2}{\|\theta_t^* - \theta_{t+M}^*\|^2}. \tag{11}$$

## B   Justification for Modified Reverse Process in Distillation

Our core modification to the reverse diffusion step is crucial for improving the gradient flow back to the initial latent $\mathcal{Z}$, allowing end-to-end optimization. A natural question arises regarding the validity of this modified process, as it intentionally deviates from standard diffusion sampling procedures designed for high-fidelity image generation that perfectly match the learned data distribution.

**Different Objectives: Distillation vs. Faithful Sampling.** The key insight is that the objective of dataset distillation differs fundamentally from standard image generation. Distillation aims to synthesize a small set of maximally informative samples that enable efficient training of downstream models, prioritizing the encoding of essential class-discriminative features over photorealism [35, 4, 42]. Pixel-perfect adherence to the original data distribution is not necessarily required or even optimal; abstract or stylized images often yield excellent distillation performance if they capture core class characteristics effectively [5].

**Impact of Modification.** Our modification introduces a direct dependency on the initial noisy state $\mathbf{z}_T$ throughout the reverse process. While preserving the Markov property (as shown in the main paper) and the representational power of the pre-trained denoiser $f_\theta$, this change means the resulting process $p_\theta(\mathbf{z}_0|\mathbf{z}_T, \mathbf{c})$ no longer guarantees sampling exactly from the original distribution $\mu$ learned by the LDM. If applied without the corrective feedback loop of distillation optimization (e.g., for unconditional generation), this modification can lead to more abstract outputs that deviate from the expected style, as illustrated with an FFHQ-trained model in Figure 5. This deviation is expected, as the process is no longer constrained solely by the standard denoising objective. We also observe a slight reduction in sample diversity (measured by average LPIPS between generated samples: 0.386 with modification vs. 0.420 without, on ImageNette samples), likely due to the persistent influence of the fixed $\mathbf{z}_T$.

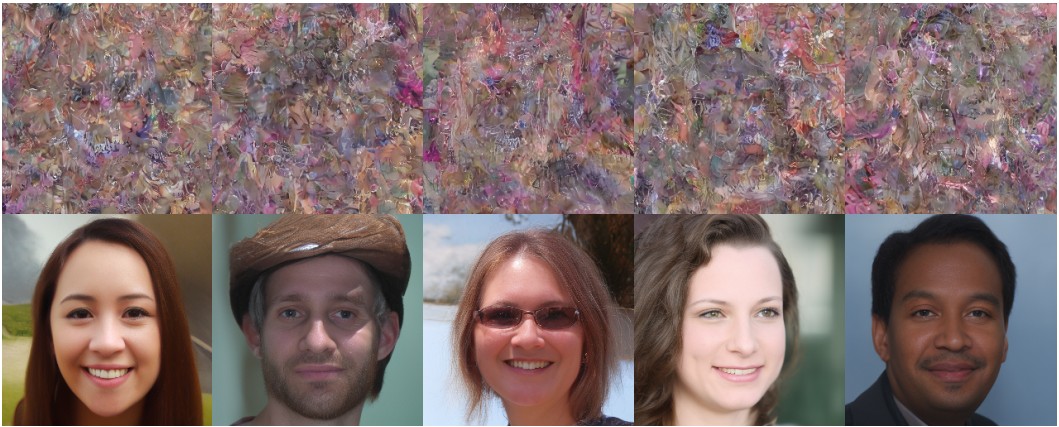

Figure 5: Influence of our modified reverse process in a classical image generation setting (unconditional FFHQ). It shows that the residual connections alter the generation process significantly, leading to abstract artifacts and the loss of coherence expected in a facial dataset: **(top)** with modification and **(bottom)** without modification.

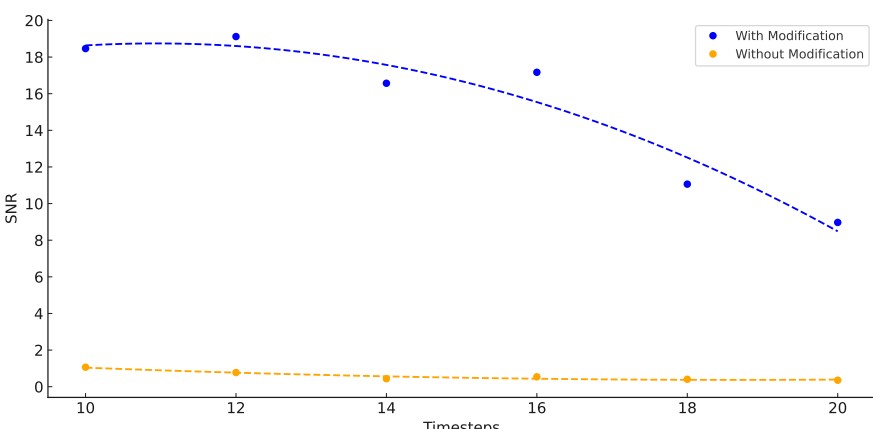

Figure 6: Gradient flow analysis comparing the Signal-to-Noise Ratio (SNR) of gradient norms for LD3M with and without our modification. Diffusion demonstrates a more stable gradient flow, indicating enhanced optimization dynamics. Dashed lines show a polyfit plot to highlight the trends.

**Suitability for Distillation.** Crucially, within the dataset distillation framework, the latent codes $\mathcal{Z}$ (and conditioning $\mathbf{c}$) are continuously optimized to minimize the distillation loss $\mathcal{L}$. This optimization process actively counteracts potential adverse effects of the modified sampling path by guiding the generation towards producing images (even if abstract) that are highly effective for the downstream task defined by $\mathcal{L}$. The essential properties needed are: (1) sufficient generative capacity to create diverse, class-relevant features, which the LDM provides; and (2) strong gradient flow for optimization, which our modification enables (as also shown in Figure 6). The empirical success of LD3M - significantly outperforming GLaD and AE-only baselines, and performing robustly even with randomly initialized LDMs - demonstrates that this trade-off (sacrificing perfect distribution matching for tractable optimization) is highly beneficial for the specific goal of dataset distillation. The resulting "abstract" representations effectively encode class information for robust generalization. Therefore, while distinct from standard sampling, our modified reverse process is a well-justified and necessary component for unlocking diffusion models for effective, end-to-end dataset distillation.

Table 9: Common hyperparameters for training the distillation algorithms used in this work.

| Parameter | Value |
|---|---|
| DSA Augmentations | Color / Crop / Cutout / Flip / Scale / Rotate |
| Iteration (Distillation) | 5,000 ($128 \times 128$) / 10,000 ($256 \times 256$) |
| Momentum | 0.5 |
| Batch Real | 256 |
| Batch Train | 256 |
| Batch Test | 128 |

## C  Hyper-Parameters for Distillation Algorithms

**LDM.** For all our LDM experiments, we set the unconditional guidance scale to the default value of 3. For $128 \times 128$ images, we used max. time steps of 10, and for $256 \times 256$ images, we used 20.

**DC.** We utilize a learning rate of $10^{-3}$ throughout our DC experiments to update the latent code representation and the conditioning information.

**DM.** In every DM experiment, we adopt a learning rate of $10^{-2}$, applying it to updates of the latent code representation alongside the conditioning information.

**MTT.** For MTT experiments, a uniform learning rate of 10 is applied to update the latent code representation and the conditioning information. We buffered 100 trajectories for expert training, each with 15 training epochs. We used ConvNet-5 and InstanceNorm. During dataset distillation, we used three expert epochs, max. start epoch of 5 and 20 synthetic steps.

## D  Large Scale Datasets

Although LD3M is compatible with various distillation algorithms -including DC, DM, and MTT -our current experiments focus on baseline variants that do not leverage inter-class relationships during optimization. This is an essential avenue for further improvement: incorporating inter-class information (e.g., through contrastive losses or hierarchical label structures) may enhance the discriminative quality of the synthetic data. Future work will explore how LD3M's expressive latent trajectories can be used to facilitate such structured, cross-class-aware distillation.

## E  Limitations

While LD3M improves dataset distillation compared to GLaD, it is essential to acknowledge certain limitations. A primary concern arises from the linear addition in the diffusion process, which may not sufficiently combat the vanishing gradient problem for larger time steps, as observed in our experiments [19]. Further alternative strategies for integrating the initial state $\mathbf{z}_T$ in the diffusion process should be evaluated to address this issue, *e.g.*, non-linear progress towards 0 as $t$ approaches 0. These alternative approaches could offer more nuanced and dynamic ways to manage the influence of $\mathbf{z}_T$ across different stages of the diffusion, potentially mitigating the problem of vanishing gradients and enhancing the overall efficacy of the distillation process.

## F  Hardware and Software

All experiments were run on a workstation equipped with an NVIDIA RTX A6000 GPU (48 GB VRAM). Our implementation uses PyTorch 1.10.1 with torchvision 0.11.2, and we build upon the GLaD library for dataset distillation with a generative prior.

Table 10: Class listings for our ImageNet subsets.

| Dataset | 0 | 1 | 2 | 3 | 4 | 5 | 6 | 7 | 8 | 9 |
|---|---|---|---|---|---|---|---|---|---|---|
| ImageNet-A | Leonberg | Probiscis Monkey | Rapeseed | Three-Toed Sloth | Cliff Dwelling | Yellow Lady's Slipper | Hamster | Gondola | Orca | Limpkin |
| ImageNet-B | Spoonbill | Website | Lorikeet | Hyena | Earthstar | Trollybus | Echidna | Pomeranian | Odometer | Ruddy Turnstone |
| ImageNet-C | Freight Car | Hummingbird | Fireboat | Disk Brake | Bee Eater | Rock Beauty | Lion | European Gallinule | Cabbage Butterfly | Goldfinch |
| ImageNet-D | Ostrich | Samoyed | Snowbird | Brabancon Griffon | Chickadee | Sorrel | Admiral | Great Gray Owl | Hornbill | Ringlet |
| ImageNet-E | Spindle | Toucan | Black Swan | King Penguin | Potter's Wheel | Photocopier | Screw | Tarantula | Sscilloscope | Lycaenid |
| ImageNette | Tench | English Springer | Cassette Player | Chainsaw | Church | French Horn | Garbage Truck | Gas Pump | Golf Ball | Parachute |
| ImageWoof | Australian Terrier | Border Terrier | Samoyed | Beagle | Shih-Tzu | English Foxhound | Rhodesian Ridgeback | Dingo | Golden Retriever | English Sheepdog |
| ImageNet-Birds | Peacock | Flamingo | Macaw | Pelican | King Penguin | Bald Eagle | Toucan | Ostrich | Black Swan | Cockatoo |
| ImageNet-Fruits | Pineapple | Banana | Strawberry | Orange | Lemon | Pomegranate | Fig | Bell Pepper | Cucumber | Granny Smith Apple |
| ImageNet-Cats | Tabby Cat | Bengal Cat | Persian Cat | Siamese Cat | Egyptian Cat | Lion | Tiger | Jaguar | Snow Leopard | Lynx |

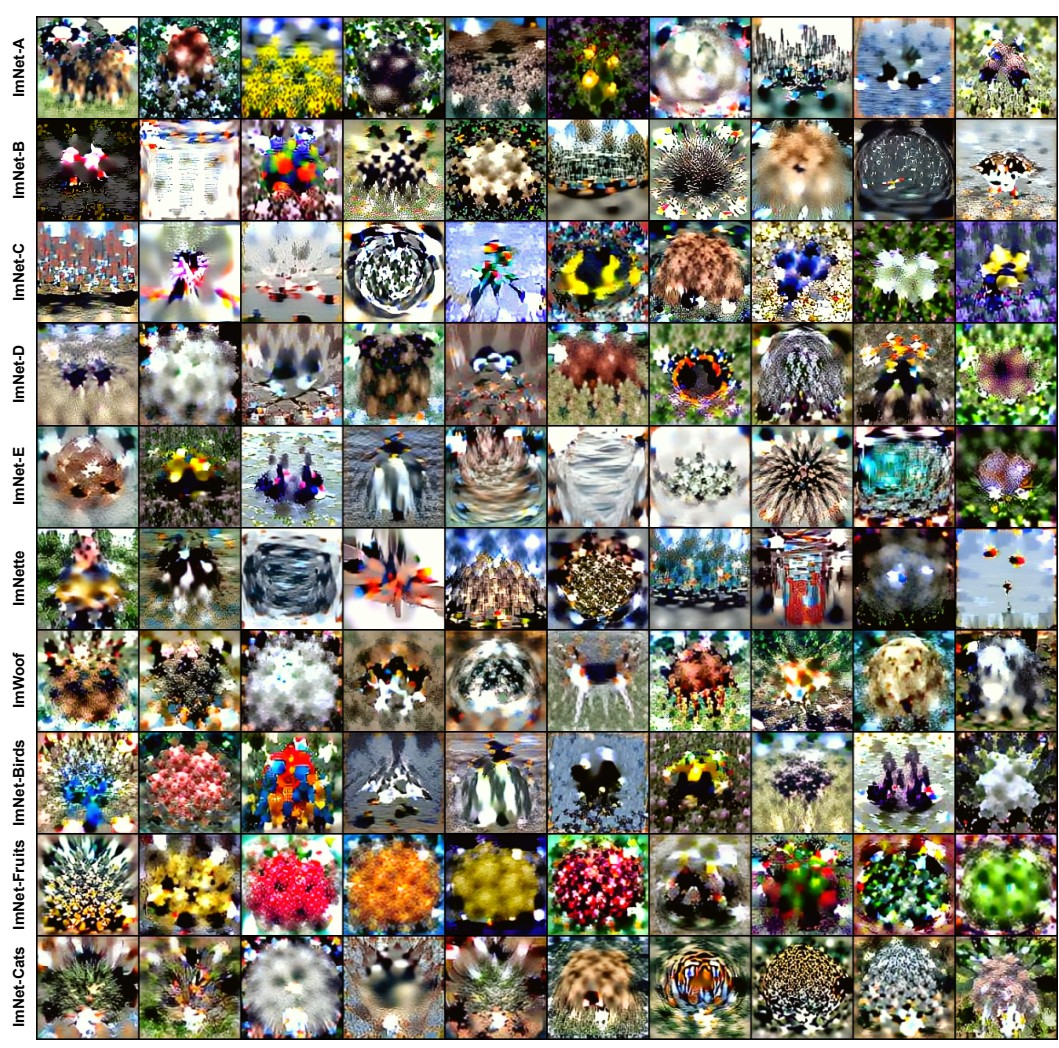

Figure 7: Images distilled by MTT in LD3M for IPC=1.

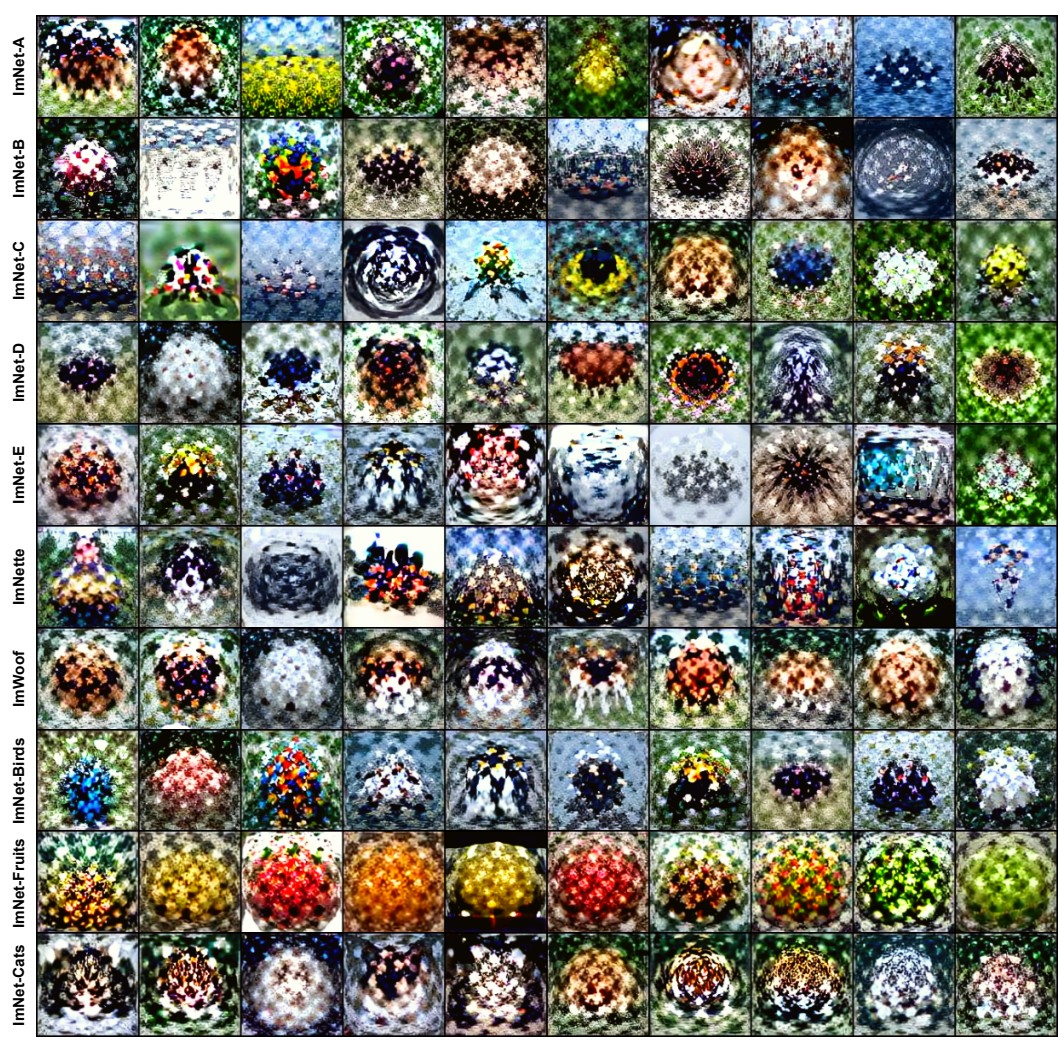

Figure 8: Images distilled by DC in LD3M for IPC=1.

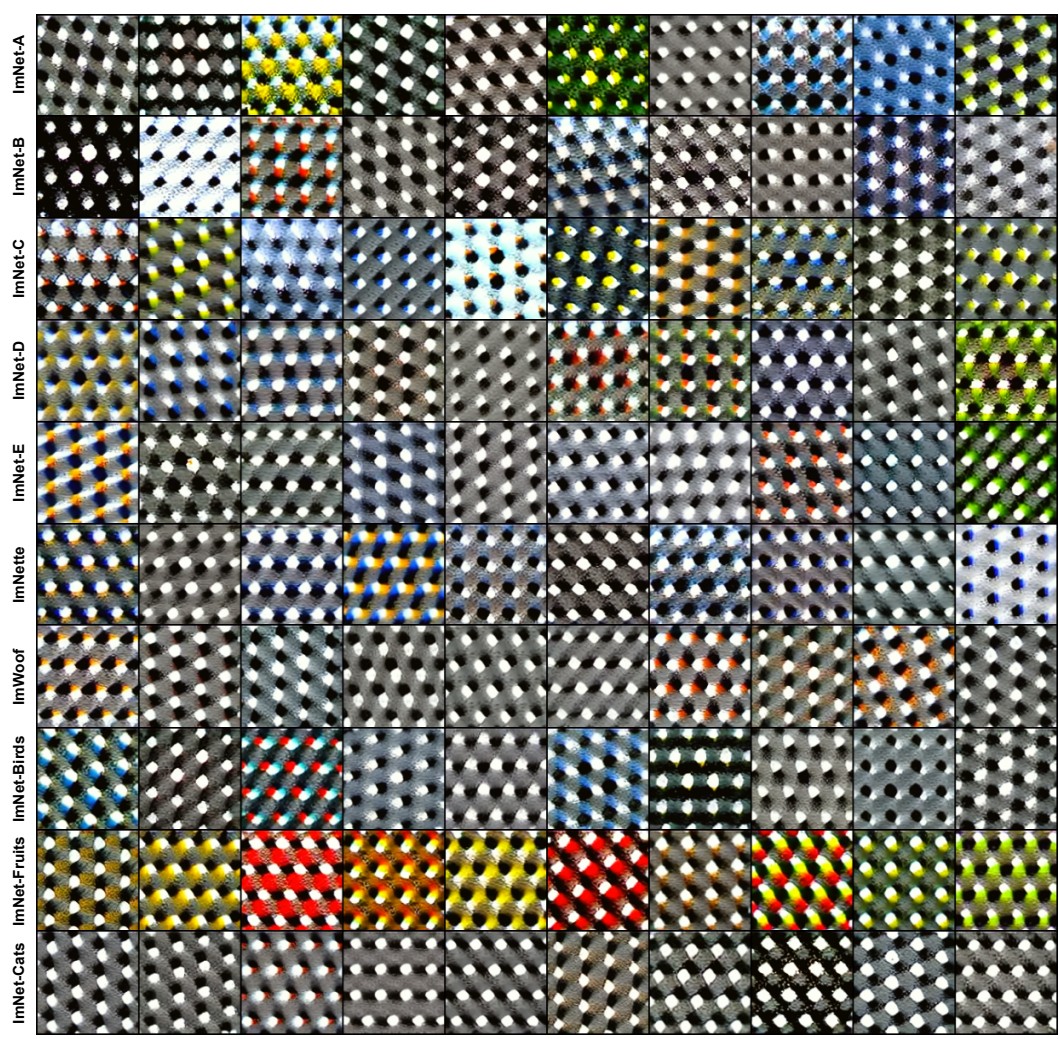

Figure 9: Images distilled by DM in LD3M for IPC=1.

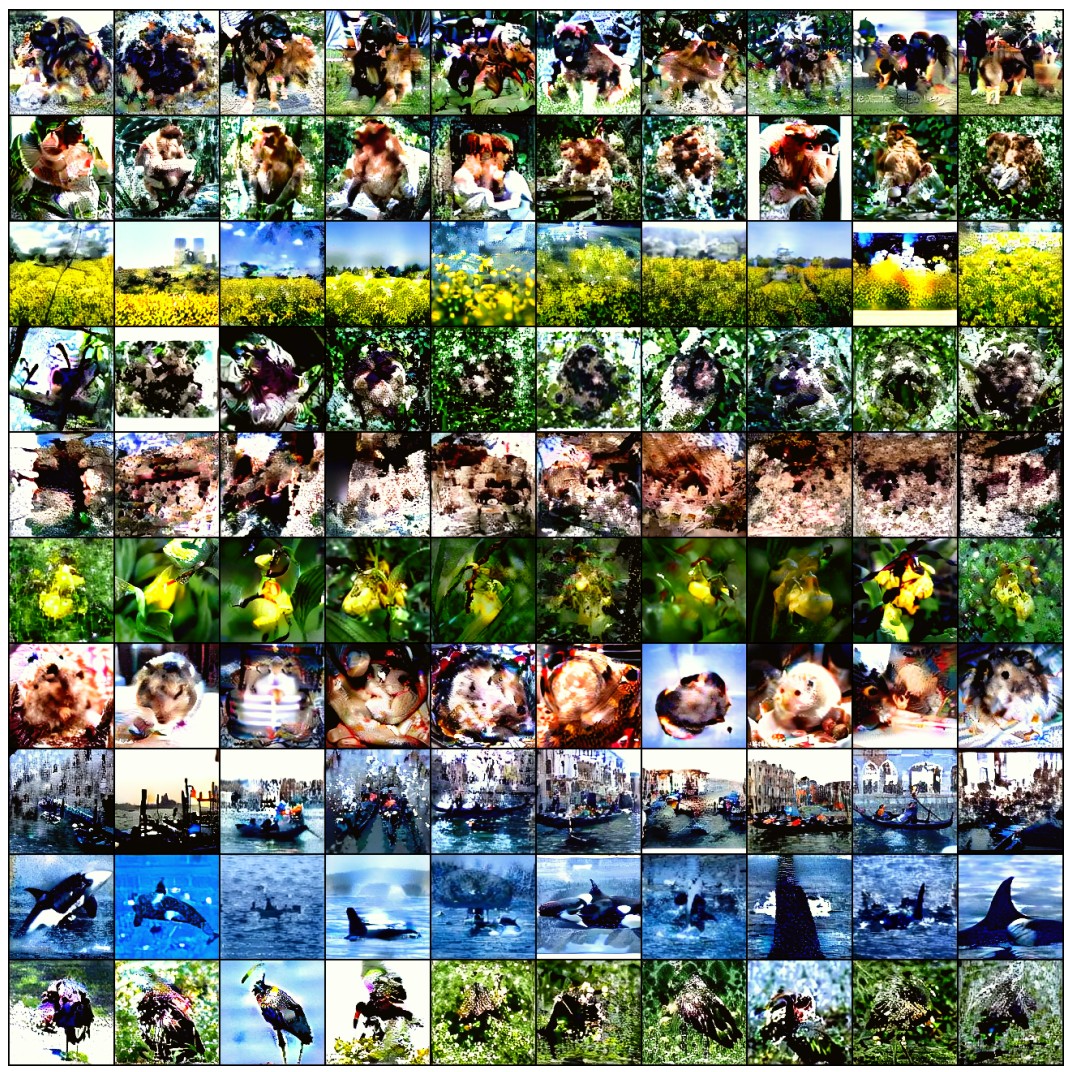

Figure 10: Images distilled by DC in LD3M for IPC=10 and ImageNet-A.

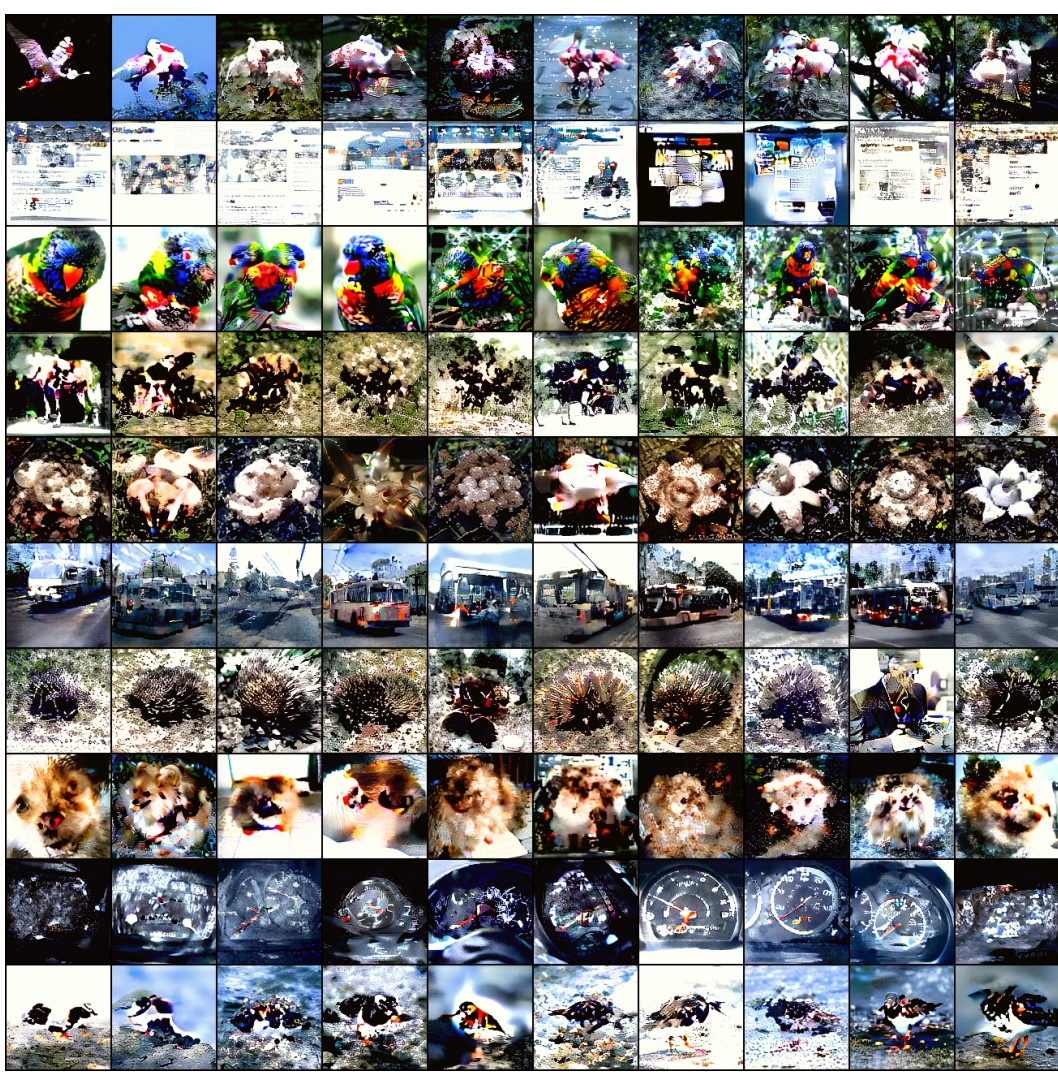

Figure 11: Images distilled by DC in LD3M for IPC=10 and ImageNet-B.

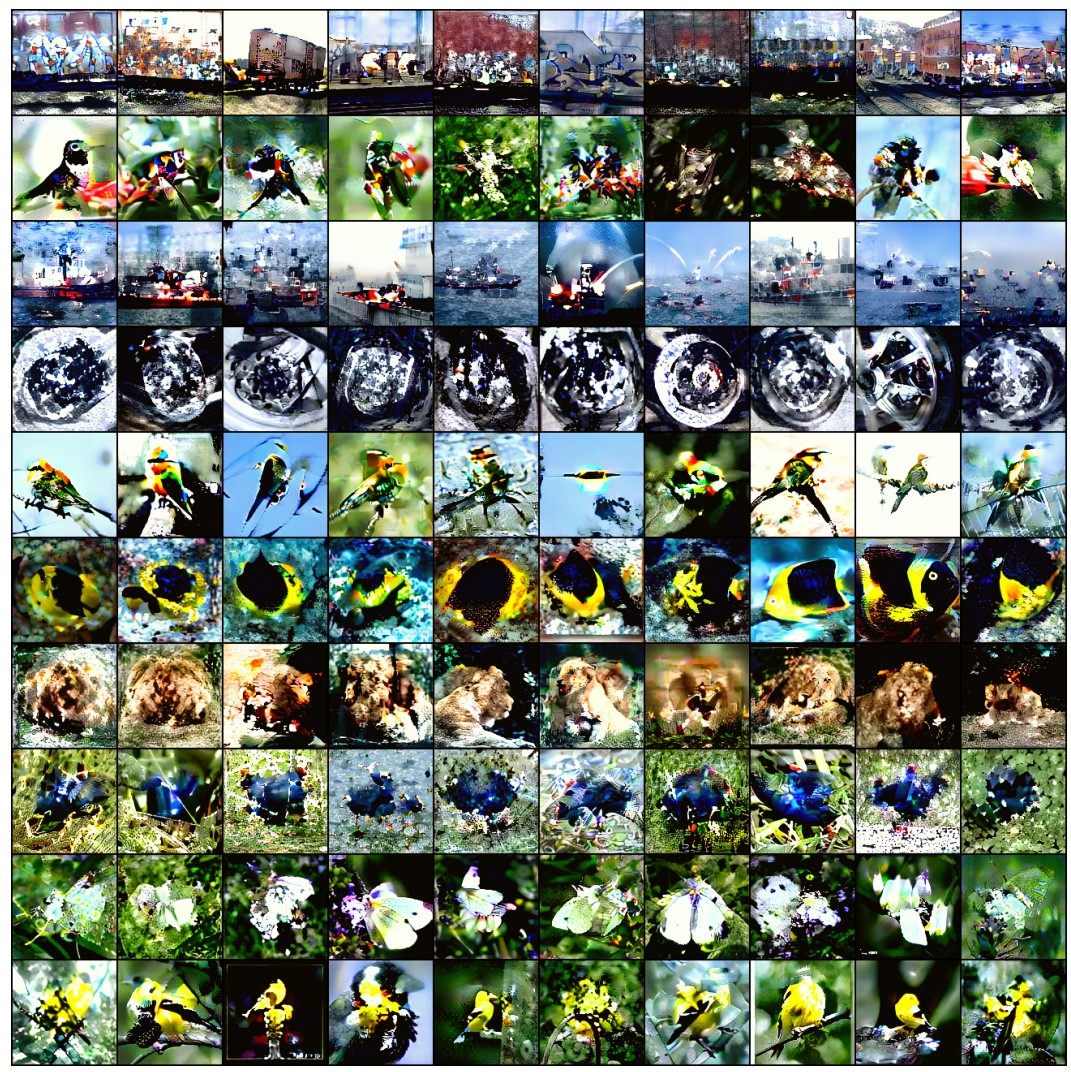

Figure 12: Images distilled by DC in LD3M for IPC=10 and ImageNet-C.

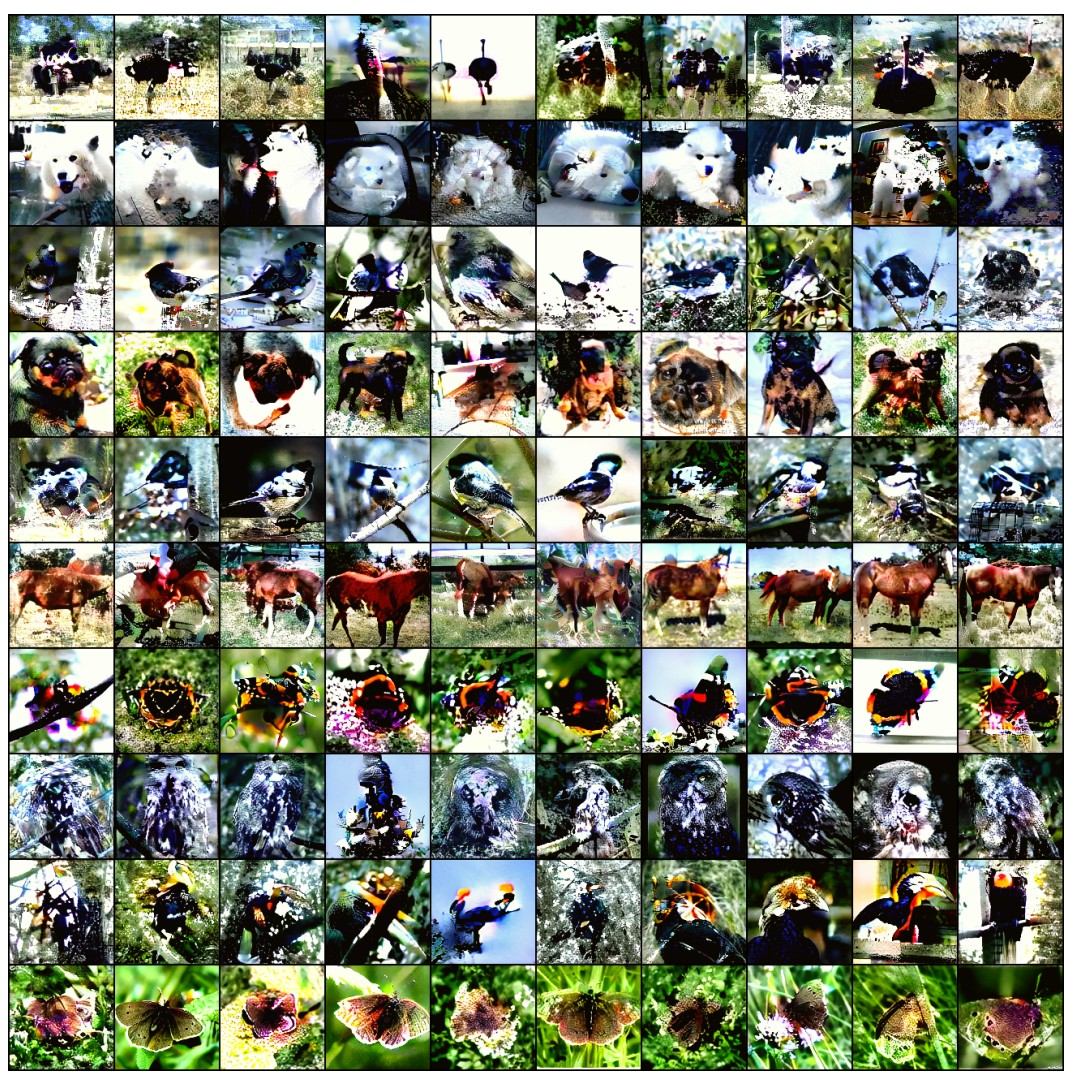

Figure 13: Images distilled by DC in LD3M for IPC=10 and ImageNet-D.

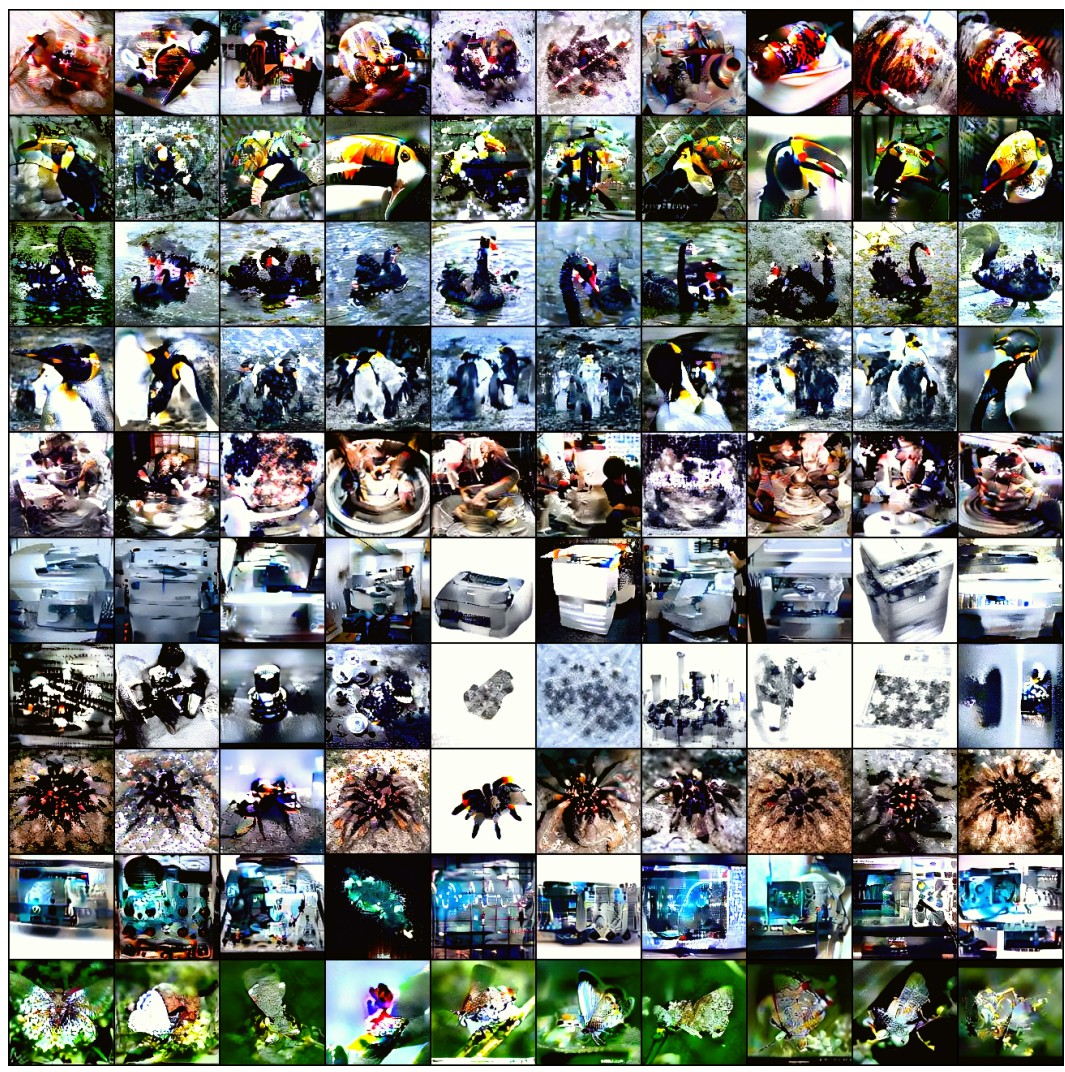

Figure 14: Images distilled by DC in LD3M for IPC=10 and ImageNet-E.

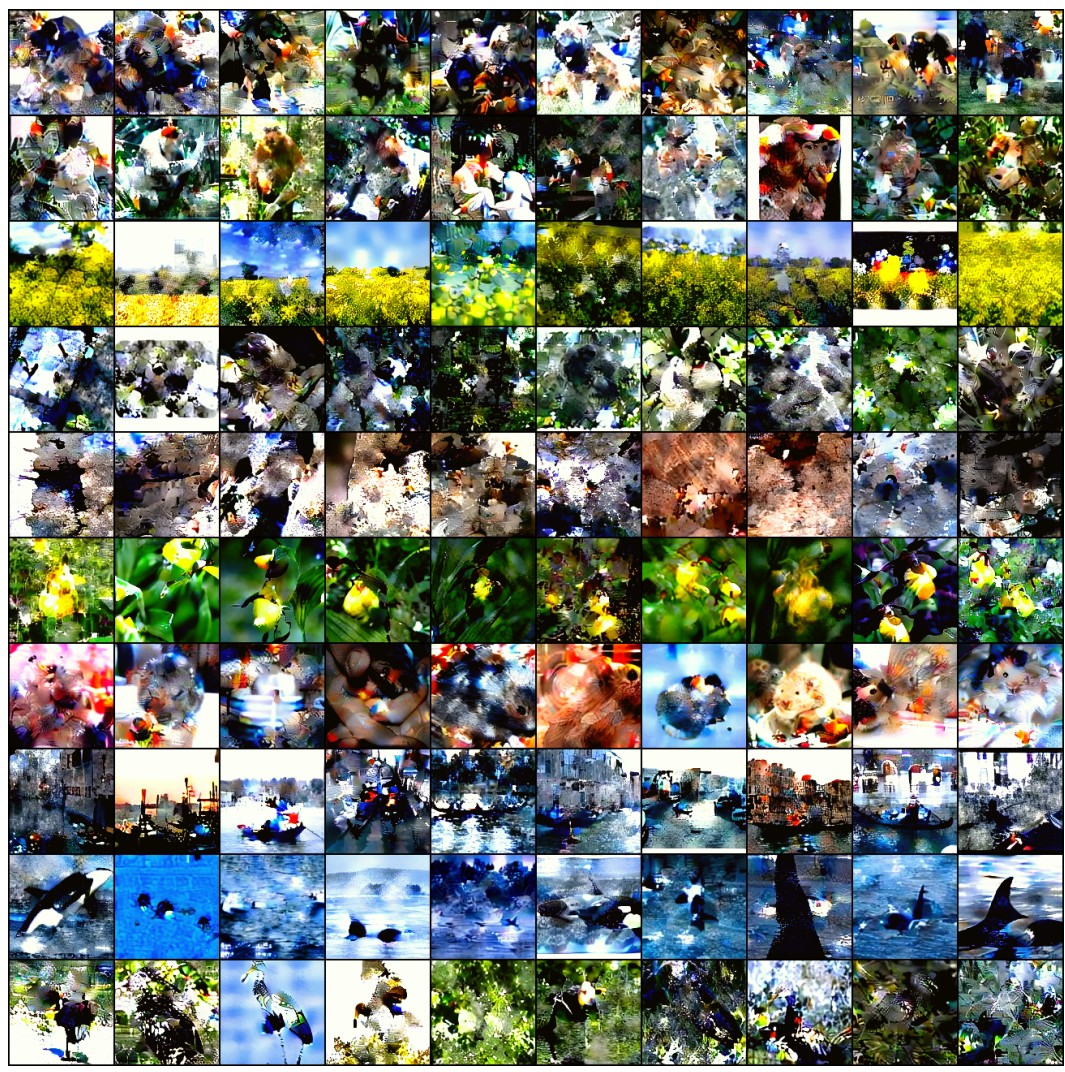

Figure 15: Images distilled by DM in LD3M for IPC=10 and ImageNet-A.

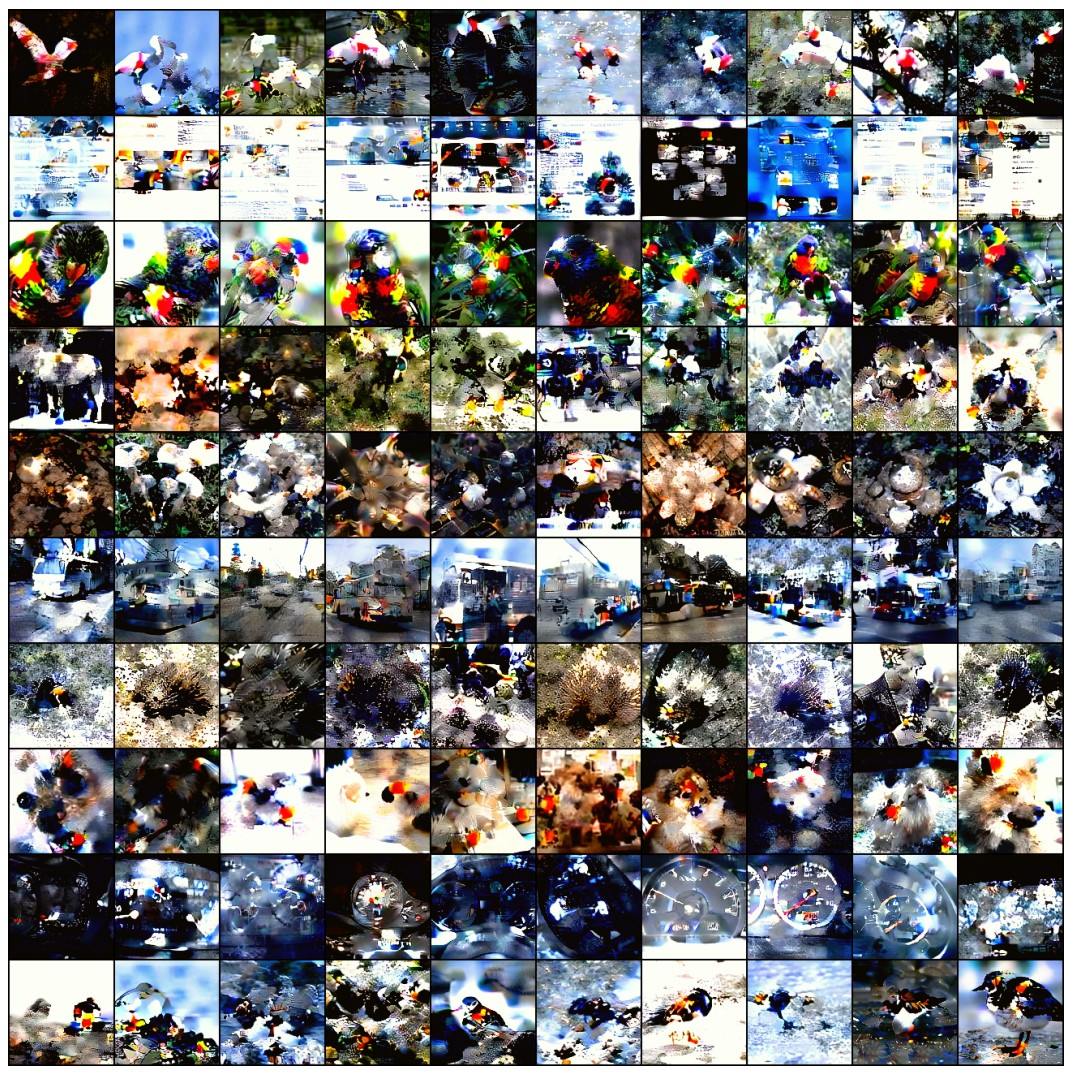

Figure 16: Images distilled by DM in LD3M for IPC=10 and ImageNet-B.

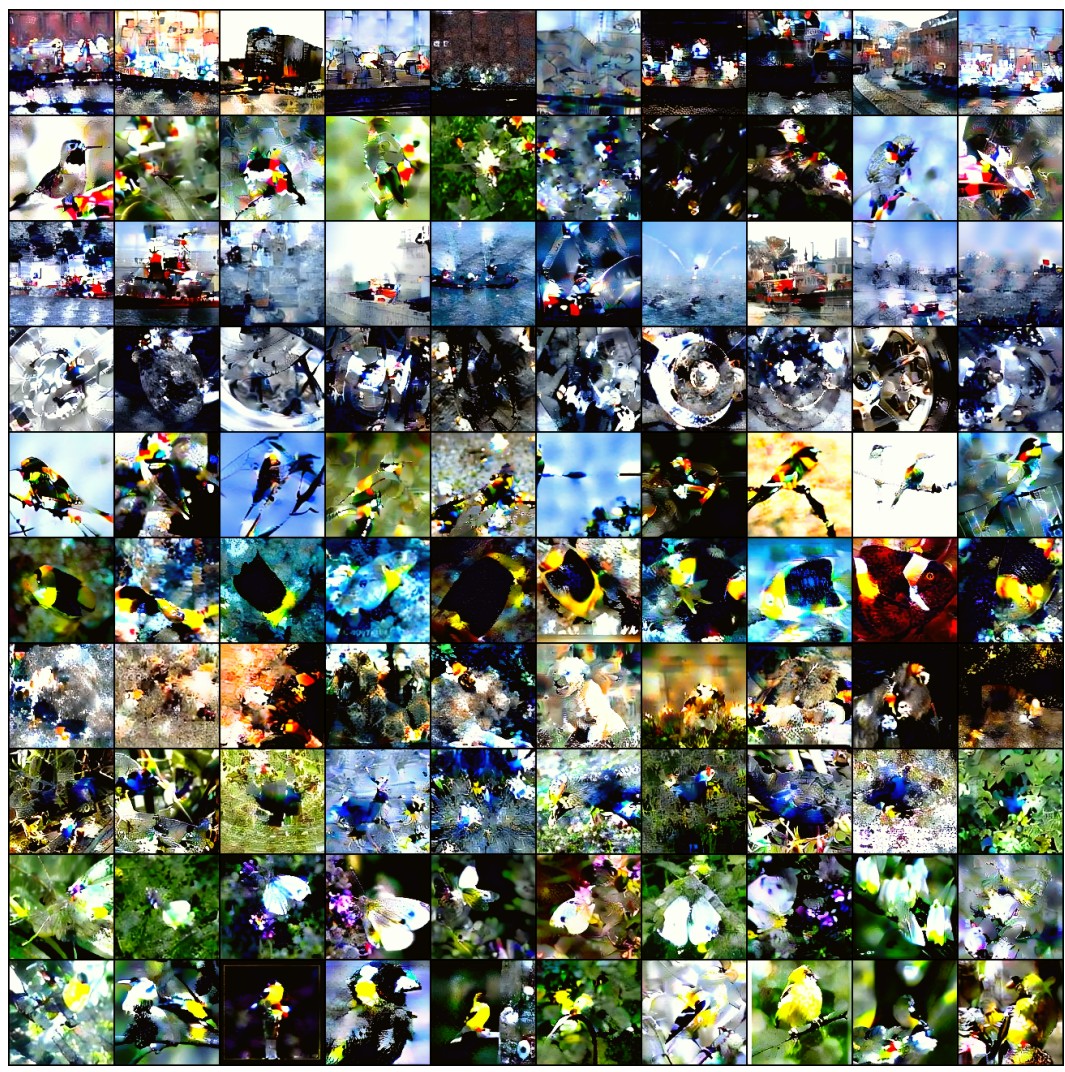

Figure 17: Images distilled by DM in LD3M for IPC=10 and ImageNet-C.

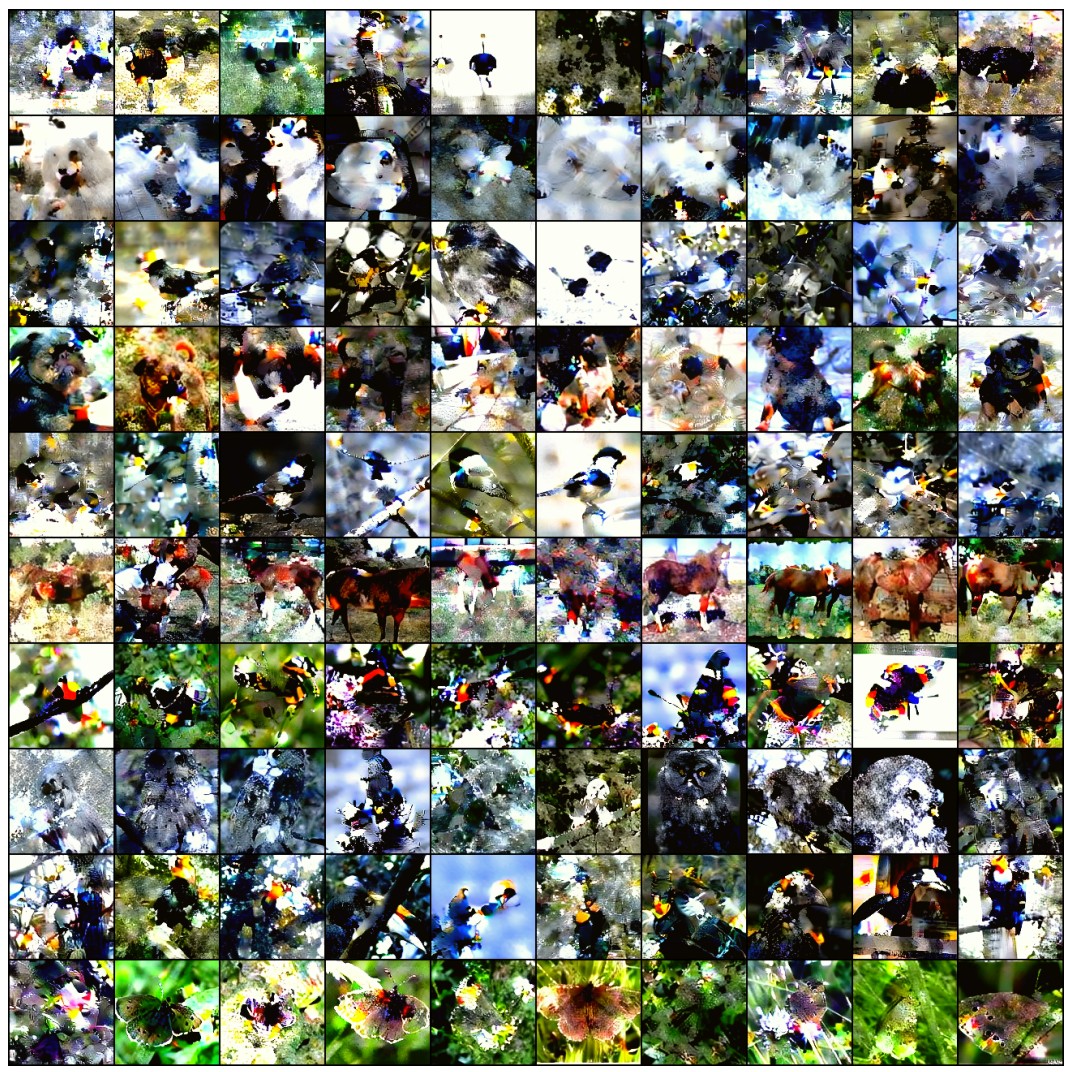

Figure 18: Images distilled by DM in LD3M for IPC=10 and ImageNet-D.

