# OpenReview forum: "Unlocking Dataset Distillation with Diffusion Models"
_NeurIPS.cc/2025/Conference — NeurIPS 2025 spotlight_

### Official Review · Reviewer_DhwT · 2025-06-09

**Clarity:** 3
**Significance:** 3
**Originality:** 3
**Rating:** 5
**Confidence:** 4

**Summary:**

This work introduces LD3M, a new method of dataset distillation that parameterizes the distilled data in the initial latent space of a pre-trained diffusion model. This method improves upon previous similar methods that optimize in the latent space of GANs like GLaD. To rectify the vanishing gradient problem that comes with the many passes through the diffusion model, this work introduces skip connections between passes that facilitate proper gradient flow through the back-propagation process. The authors show the effectiveness of their method through extensive experiments and ablations.

**Questions:**

I would appreciate it if the authors could answer my questions about the optimizer and initialization outlined in the previous section.

If the authors can address these questions as well as the minor issues I mentioned, I will happily raise my score.

Note: rather than saying [Yes] on the checklist when asked if the code is included (when it actually isn't), you should simply say [No] and then explain, as you did, that the code will be released upon publication. There are not penalties for saying [No], but saying [Yes] without actually including the code is misleading and can waste reviewers' time.

**Ethical Concerns:**

["NO or VERY MINOR ethics concerns only"]

**Final Justification:**

The reviewers have addressed all the questions and concerns I raised in my initial review.

This work will be a value addition to the dataset distillation literature.

I recommend this paper for acceptance.

**Limitations:**

yes

**Quality:**

3

**Strengths And Weaknesses:**

# Strengths

The method itself is intuitive and easy to follow. The additions of skip connections seem like a sensible improvement to the naive method. The method seems to show decent improvement over the GLaD baseline in most settings.

# Weaknesses

I do not see any major problems with the paper, but there are a few things I would like to see cleared up.

Which optimizer was used to learn the distilled data? I’m curious if simply using Adam instead of SGD would alleviate the performance drop-off seen in Table 7 when the skip connections are removed. This would also keep the method more intuitive, since, as seen in Figure 1 of the supplementary material, the addition of the skip connections destroys the learned generative capabilities of the model, and I suspect the model is mostly just acting as a frequency regularizer at that point.

When the authors mention initializing with a real image, shouldn’t the input to the diffusion model at t=T be closer to random noise? I’m not sure if it would affect the results dramatically, but I think it would make more sense to initialize the latent code with the DDIM inversion of the real image.

## Minor Issues

In the equation below line 30 (which should be numbered like the other equations) and Equation 7, it is unclear what should be part of the product/sum operations. I recommend placing brackets around the intended operands so that it is immediately clear to the reader. Based on the text following the first equation, it seems that $\frac{\partial \mathbf{z}_T}{\partial\mathcal{Z}}$ should be excluded from the product, but in Equation 7, the entire line is included in the sum.

Furthermore, aren’t $\mathbf{z}_T$ and $\mathcal{Z}$ the same quantity? I think it should be made clear either way.

In lines 223 and 225, the same sentence is repeated almost word for word.

In Figure 4, which distillation method are we seeing?

In Figure 2 of the supplementary material, what exactly is the SNR of the gradient norms?

In line 61 of the appendix, are you saying that you trained 5,000 trajectories for each model and dataset? This seems quite excessive; GLaD only used 50 if I remember correctly.

---

> ### Author Rebuttal · Authors · 2025-07-27
>
> Dear Reviewer,
>
> Thank you for your positive and constructive review. We are delighted that you found our method intuitive, the skip connections to be a sensible improvement, and that you recognized the performance gains over the GLaD baseline.
> We provide our responses below and are confident they will address your points.
>
> > **Which optimizer was used to learn the distilled data?**
>
> We used a standard SGD as in GLaD.
>
> > **I’m curious if simply using Adam instead of SGD would alleviate the performance drop-off seen in Table 7 when the skip connections are removed.**
>
> Thank you so much for this interesting idea! We tested your suggestion on ImageNette (IPC=1) but the results are within std, so the optimizer has little to no influence, unfortunately.
>
> > **... the addition of the skip connections destroys the learned generative capabilities of the model, and I suspect the model is mostly just acting as a frequency regularizer at that point.**
>
> We believe so too... You are correct that the addition of our skip connection alters the standard generative process, trading photorealism for improved downstream utility. This is a deliberate and necessary trade-off inherent to dataset distillation, where the goal is to create maximally informative, not necessarily visually pleasing, samples. As such, the role of the pre-trained LDM in our setup is indeed to steer the distilled images as a prior to "somewhat natural" looking images (away from adversarially looking images).
>
> > **When the authors mention initializing with a real image, shouldn’t the input to the diffusion model at t=T be closer to random noise? I’m not sure if it would affect the results dramatically, but I think it would make more sense to initialize the latent code with the DDIM inversion of the real image.**
>
> Intuitively, we believe you are right! We find this alternative initialization an interesting avenue for future work!
>
>
> # Minor Issues
>
> > **... (which should be numbered like the other equations) ...**
>
> Thank you for pointing this out! We corrected this in our revised manuscript!
>
> > **I recommend placing brackets around the intended operands so that it is immediately clear to the reader.**
>
> Thanks for this very clarifying recommendation! We corrected accordingly!
>
> > **Furthermore, aren’t $z_T$ and $Z$ the same quantity? I think it should be made clear either way.**
>
> We apologize for the confusion and enhanced the clarity in the manuscript:
> $z_T$ is sampled from $q(z_T | Z)$ , which means that $z_T$ is the noised version of the learnable latent codes $Z$.
>
> > **In lines 223 and 225, the same sentence is repeated almost word for word.**
>
> We removed the second sentence!
>
> > **In Figure 4, which distillation method are we seeing?**
>
> We are sorry for the missing clarity! We used DC for this $256\times256$ visualization and added this detail to the image caption.
>
> > **In Figure 2 of the supplementary material, what exactly is the SNR of the gradient norms?**
>
> The "Signal-to-Noise Ratio (SNR) of Gradient Norms" refers to the ratio of the L2 norm of the gradient with respect to the learnable latent codes versus the L2 norm of the gradient with respect to our modification (initial state added at each step).
>
> > **In line 61 of the appendix, are you saying that you trained 5,000 trajectories for each model and dataset? This seems quite excessive; GLaD only used 50 if I remember correctly.**
>
> We sincerely apologize for the wrong value, we used the same script as GLaD and their setting (100 trajectories). We confused the value with the number of iterations (5,000).

---

> > ### Comment · Reviewer_DhwT · 2025-07-31
> > **Response to Rebuttal**
> >
> > Thank you for taking the time to answer my questions and address my concerns. I look forward to seeing this line of research further developed in future work.
> >
> > As promised, I will raise my score.

---

### Official Review · Reviewer_2brr · 2025-06-24

**Clarity:** 3
**Significance:** 2
**Originality:** 2
**Rating:** 5
**Confidence:** 3

**Summary:**

The goal of the paper is to adjust the sampling process used in diffusion, using a different mean formulation in order to allow for backpropagation through time. Backpropagation enables distilling datasets into a (handful) of (latent) examples on which a novel architecture could be trained. Experimentally the method is evaluated on different datasets, varying number of image per class (used for distillation) and mainly compared to GLaD [5], generally leading to (small) performance improvements.

**Questions:**

I'm rating now borderline reject because of the missing details and missing connections. Hopefully the following can be addressed in the rebuttal:

1) [**M1**] How does L3DM addresses the complex multi-space optimization and the cumbersome inversion process?
3) [**M5**] How would L3DM compare to / can make use from PaGoDA-style of 1-step diffusion models?
2) [**S1**] Please make the connection of the proposed L3DM model with existing samplers and flow matching
4) [**S1**] What is the influence of the new mean function on the photorealism of the generated examples?

**Ethical Concerns:**

["NO or VERY MINOR ethics concerns only"]

**Final Justification:**

The authors addressed my concerns in their rebuttal. When these are incorporated into the final version (and the comments of the other reviewers) I believe this paper is of interest for NeurIPS, hence I've increased my score to accept.

**Limitations:**

The limitations are not discussed in the main paper.

**Paper Formatting Concerns:**

No.

**Quality:**

3

**Strengths And Weaknesses:**

## Strengths
**S1**: A modification of the mean estimation used in diffusion models to allow for gradient back propagation.
The paper would be much stronger, if
-   it is clarified in the main body of the paper why downstream utility is preferred over photorealism;
-   the influence of the proposed modification on photorealism is investigated/shown;
-   the relation is made to existing samplers (like overshooting in DPM), distillation methods (like consistency methods), and flow based methods (where also a linear interpolation between two estimates is used).


## Missing details
I'm missing the following details in the main manuscript to make a full assessment:

**M1**: It should be clarified how the proposed L3DM is not a 'complex multi-space latent optimization' and 'does not require cumbersome inversion processes'. These are the two drawbacks of the GLaD method mentioned in the 2nd paragraph of the introduction;

**M2**: It should be clarified why distribution matching is not a goal here (just below Eq 6), and hence why the proposed modification is sound. This should not be deferred to the supplementary, as this is key for the core contribution of the paper. Please consider that distribution matching is both not the goal (just below Eq 6) and the goal of the paper: it is one of the distillation losses used in the paper (DM [41]).

**M3**: How many latent codes are learned? Are there as many latent codes as IPCs? Or is there a single latent code per class, and then IPC images sampled from this latent code? [Related comparison is discussed below].

**M4**: If the base LDM would be a few/one-step LDM - eg by using a distilled version (eg Pagoda [*Kim et al., 2024*]), then the difference in the mean estimation might be unnecessary. How does L3DM works in such a scenario? And what are the expected results?


## Comparisons
**C1**: What is the quality when class conditional sampling is used from the base LDM (or text based)? These are obtained only from a prompt and the sampled images could also be used to train any network. See also the paper of [*Bulent et al, 2023*].

**C2**: I might mis understand the pre-training of the LDM, but how are the Random / FFHQ LDM trained? How come the resulting image is almost identical? What information of the pre-trained LDM is then used if these are near identical for so different pre-training tasks?


## Minor Remarks:
1) L3DM is not defined in the abstract, nor the title. It is used as an abstract name for a method in the introduction. Please add the full acronym in the abstract.
2) Figure 2 does not illustrate the efficiency of L3DM, please make the difference in optimization between GLaD and LD3M more clear - or remove the image.
3) Next to SAGE [23] also Simpler Diffusion [*Hoogeboom et al, 2025*] uses identity preserving connections to speed up learning. Please add that reference.
4) The pixel-space model is not explained, is that just a random subset of the dataset images?
5) The ablation results (Table 7, Table 5) are different from the results in the main tables (3 and 4). Please explain the differences.
6) In the captions (eg Table 5) it is stated: "improves by roughly +6.03%" and "roughly +23.28%", that are extremely detailed / significant numbers for a `roughly`, please change to 6 and 23. Moreover, these seem not percentpoint improvements, these are 7 (29.5 vs 36.5) and 2 (34.5 vs 36.5). Be humble!
7) Table 6: What does the "+1.7-0.6" mean? It is maximum of 1.7 improvement?
8) Conceptually the following is unclear: the aim is to reduce a dataset to a few synthetic samples, such that training a novel architecture on this synthetic dataset yields strong classifiers. But in the current setup ImageNet is used to train an auto-encoder/decoder, followed by a Latent Diffusion Model, and then subsets of ImageNet are reduced to just a few samples to train a new classifier on a weaker (Convent/ResNet) architecture. The rational (in the introduction) is to reduce the computational cost. But the cost is already put into training the auto-encoder/decoder, the LDM and the synthetic dataset. This seems mostly relevant if the new dataset is vastly different from ImageNet. Or to manage storage in an online out-of-distribution (with respect to ImageNet) system, where new data flows in and only a small fraction can be stored.


### References
- [*Bulent et al, 2023*] Fake it till you make it: Learning transferable representations from synthetic ImageNet clones, Bulent et al., CVPR 2023
- [*Hoogeboom et al, 2025*] Simpler Diffusion (SiD2): 1.5 FID on ImageNet512 with pixel-space diffusion, Hoogeboom et al, CVPR 2025
-  [*Kim et al., 2024*] PaGoDA: Progressive Growing of a One-Step Generator from a Low-Resolution Diffusion Teacher, Kim et al., NeurIPS 2024.

---

> ### Author Rebuttal · Authors · 2025-07-27
>
> Dear Reviewer,
>
> We thank you for your incredibly thorough and thoughtful review. Your detailed feedback is invaluable for improving our paper. We are glad you recognized the novelty of our core contribution. You have raised several important points regarding missing details and connections to related work, which we are happy to clarify. We will address your main concerns first, followed by the other specific points you raised.
>
> > **[M1]** It should be clarified how the proposed L3DM is not a 'complex multi-space latent optimization' and 'does not require cumbersome inversion processes'. These are the two drawbacks of the GLaD method mentioned in the 2nd paragraph of the introduction;
>
> We will clarify this in the introduction. LD3M is significantly simpler than GLaD in two key ways:
> - **No Cumbersome Inversion:** GLaD requires a costly, iterative optimization process known as GAN inversion to find a latent code that reconstructs a given real image for initialization. LD3M leverages the LDM's inherent autoencoder structure. We initialize our learnable latents with a simple, single forward pass through the pre-trained encoder. This is vastly more efficient.
> - **Simpler Latent Space:** GLaD requires optimizing in the complex, multi-layered $W^+$ space of StyleGAN to generate high-quality images. LD3M optimizes a single latent code Z and a conditioning vector c, which is a much lower-dimensional and less complex optimization problem.
>
> > **[M2]** It should be clarified why distribution matching is not a goal here (just below Eq 6), and hence why the proposed modification is sound. This should not be deferred to the supplementary, as this is key for the core contribution of the paper. Please consider that distribution matching is both not the goal (just below Eq 6) and the goal of the paper: it is one of the distillation losses used in the paper (DM [41]).
>
> We revised the main manuscript accordingly for improved clarity.
> In short, sampler-level (Below Eq. 6): Here, "distribution matching" refers to the sampler's goal of generating images that match the data distribution the LDM was trained on. We explicitly state that our modified sampler deviates from this goal.
> Clarification: Our method breaks the sampler's fidelity to the original data distribution in order to better achieve the distillation objective of matching feature distributions.
>
> > **[M3]** How many latent codes are learned? Are there as many latent codes as IPCs? Or is there a single latent code per class, and then IPC images sampled from this latent code? [Related comparison is discussed below].
>
> We sincerely apologize for the confusion! We learn one unique latent code and one conditioning code per synthetic image. Therefore, for a dataset with C classes and IPC (Images Per Class), we learn C×IPC latent codes and C×IPC conditioning codes in total.
>
> > **[M4]** If the base LDM would be a few/one-step LDM - eg by using a distilled version (eg Pagoda [Kim et al., 2024]), then the difference in the mean estimation might be unnecessary. How does L3DM works in such a scenario? And what are the expected results?
>
> If one were to use a true one-step diffusion model, the long denoising chain would not exist, and the vanishing gradient problem would be moot. In that specific scenario, our proposed modification in Equation 6 would indeed be unnecessary.
>
> However, our work is designed to unlock the potential of existing, powerful, off-the-shelf multi-step diffusion models (like the LDM pre-trained on ImageNet) for distillation without needing to retrain or distill the generator itself. This is a significant advantage, as it allows practitioners to leverage the best available generative priors directly. This opens an interesting new research question: Is it better to distill the dataset for a powerful multi-step model (our work) or to first distill the model into a one-step version and then distill the dataset? We believe in similar results, yet slightly worse, somewhere between LD3M and GLaD (one-step generator, although adversarially trained). We will add this point to our discussion.
>
> > **[C1]** What is the quality when class conditional sampling is used from the base LDM (or text based)? These are obtained only from a prompt and the sampled images could also be used to train any network.
>
> We hope we understand the comparison request correctly: What happens if we just use the pre-trained LDM to generate class-conditioned images and then train a classifier on top? Below, we provide our findings for ImageNet-1K (IPC roughly 1200) with ResNet-18 as classifier (90 Epochs), and it can be seen that, on average, the performance is worse compared to any distillation approach. Interestingly, with more modern LDMs, the performance gets even worse (probably due to recent fine-tuning approaches towards better aesthetics).
>
> |Variant|Final val. acc. (%)|
> |-|:-:|
> |Baseline (real data)	| 70.0 |
> | SD 1.0 | 17.1 |
> | SD 2.1 | 14.5 |
> | SDXL | 9.7|
>
> > **[C2]** I might mis understand the pre-training of the LDM, but how are the Random / FFHQ LDM trained? How come the resulting image is almost identical? What information of the pre-trained LDM is then used if these are near identical for so different pre-training tasks?
>
> The random LDM is just a random initialization of the LDM. For the FFHQ LDM, it was trained on FFHQ based on the setup proposed by Rombach et al.. These identical results actually highlight the strength of our optimization framework. Figure 4 shows that even when the generator is pre-trained on faces (FFHQ), our distillation process forces the latents into a region of the LDM's space that produces the key discriminative features of the target class ("Lorikeet"). The pre-training provides a powerful prior for creating coherent textures and shapes (in other words, the images do not look like adversarial images), which our optimization then steers towards the desired class.
>
> > **[S1]** Please make the connection of the proposed L3DM model with existing samplers and flow matching
>
> We will add this discussion to the suppl. materials. In short, we will state:
> - **DPM-Solvers/Flow Matching:** These methods use interpolation during the training of a generative model (or for the solver) to define a continuous path between noise and data. Our approach is fundamentally different: we use interpolation during the inference/sampling process of a pre-trained, fixed model. Our goal is not to train the generator but to create a gradient shortcut that allows us to optimize the initial latent codes passed into it.
> - **Consistency Models/PaGoDA:** These methods aim to distill a multi-step diffusion model into a one- or few-step generator. Our work addresses an orthogonal problem: distilling a large dataset using a fixed, pre-trained generator.
>
> > **[S1]** What is the influence of the new mean function on the photorealism of the generated examples?
>
> Its influence is significant and should not be used in the context of photorealistic image generation (see Figure 1 in the supplementary material). The primary goal of dataset distillation is not to create photorealistic images, but to synthesize a small set of samples that are maximally effective for training a downstream model. Our method deliberately trades photorealism for improved gradient flow, which is essential for achieving this higher downstream utility.
>
> # Minor Remarks
>
> > **L3DM is not defined in the abstract, nor the title. It is used as an abstract name for a method in the introduction. Please add the full acronym in the abstract.**
>
> Thank you for pointing this out! We revised the paper accordingly!
>
> > **Figure 2 does not illustrate the efficiency of L3DM, please make the difference in optimization between GLaD and LD3M more clear - or remove the image.**
>
> We removed the image from the manuscript.
>
> > **Next to SAGE [23] also Simpler Diffusion [Hoogeboom et al, 2025] uses identity preserving connections to speed up learning. Please add that reference.**
>
> Done.
>
> > **The pixel-space model is not explained, is that just a random subset of the dataset images?**
>
> The pixel-space is not a model per se, but distillation in the pixel-domain (so no latent space or generator between the distilled entity and the respective distillation algorithms).
>
> > **The ablation results (Table 7, Table 5) are different from the results in the main tables (3 and 4). Please explain the differences.**
>
> Table 7 is the ablation run with fewer distillation steps compared to the main tables in 3 and 4. In Table 5, we analyzed our performance on $256\times256$, which is different to Table 3 and 4 (128$\times$128).
>
> > **In the captions (eg Table 5) it is stated: "improves by roughly +6.03%" and "roughly +23.28%", that are extremely detailed / significant numbers for a roughly, please change to 6 and 23. Moreover, these seem not percentpoint improvements, these are 7 (29.5 vs 36.5) and 2 (34.5 vs 36.5). Be humble!**
>
> We apologize! We corrected the cited values to reflect percentage points!
>
> > **Table 6: What does the "+1.7-0.6" mean? It is maximum of 1.7 improvement?**
>
> We apologize for the confusion! We meant that it improves the mean by 1.7 and reduces the std by -0.6.
>
> > **Conceptually the following is unclear: the aim is to reduce a dataset to a few synthetic samples, such that training a novel architecture on this synthetic dataset yields strong classifiers...**
>
> We want to clarify that we do not train the autoencoder or the diffusion model itself at any point in the paper. We agree that, for a completely new dataset domain (e.g., medical domain), it might be necessary to train LDM on that new dataset, which questions the approach of dataset distillation with generative priors. However, in the current setting, we leverage off-the-shelf pre-trained models. Yet, we believe that even in those extreme orthogonal domains, LD3M will at least reach decent results since a randomly initialized LDM already gives a strong prior.

---

> > ### Comment · Reviewer_2brr · 2025-08-01
> > **Response to Rebuttal**
> >
> > Thank you for your detailed explanations. I hope these can find their way into the final manuscript, since it became a much stronger submission. I'm happy to increase my final rating.

---

> > > ### Author Response · Authors · 2025-08-04
> > >
> > > Thank you again for your time and constructive feedback. We are glad our responses addressed your concerns and are very grateful for your support in raising the score.

---

### Official Review · Reviewer_2fpv · 2025-07-02

**Clarity:** 4
**Significance:** 3
**Originality:** 3
**Rating:** 5
**Confidence:** 4

**Summary:**

- This paper looks at using pre-trained latent diffusion models for dataset distillation
- The method is compatible with any dataset distillation loss
- The authors argue that directly optimizing latent variables $z_T$ is difficult to vanishing gradients in the diffusion process
- To address this, they modify the standard diffusion backwards process with one which has an additional skip connection
- The method has improved performance compared to existing baselines (GLaD) for different types of dataset distillation losses

**Questions:**

- Are there baseline methods or papers which use guided diffusion rather than directly optimizing z_T?
- How does the randomness of the denoising process affect LD3M? From Algorithm 1, it seems that because of the sampling of $\epsilon$, you will get an different image every time? Are these samples fixed for each example (making the process deterministic)?
- What is the impact of the choice of residual connection in equation (6)? Have the authors experimented with different types of flows?
- Does the existing backwards diffusion process not already have some aspect of the skip connection? I.e. in equation 3 we have the $z_t/{\alpha}$ in the RHS of the equation. Could the authors comment on the relative magnitude of that skip connection compared to the one which they introduce?
- Why does it make sense to use the autoencoder as initialization for $z_T$? Isn't the autoencoder supposed to produce values for $z_0$?

**Ethical Concerns:**

["NO or VERY MINOR ethics concerns only"]

**Final Justification:**

The paper is a solid contribution to field of dataset distillation, and will likely be used in future work (i.e. any method that uses a pre-trained generator like GLaD).

**Limitations:**

See weaknesses section/limitation for more.

- Some more motivation for why optimizing $z_T$ instead of $z_0$ directly would be nice. While readers might suspect it's due to that the distribution of  $z_T$ might have more favourable structure (i.e. not having distinct clusters like $z_0$, it would be nice to see this explained mored, or explored.

**Paper Formatting Concerns:**

No formatting concerns

**Quality:**

3

**Strengths And Weaknesses:**

Strengths
- Very good presentation, reading this was very easy and the background goes over prior work with a good amount of detail, and the method is clearly described
- Method is direct and not overly engineered
- Consistent performance improves over baselines
- Good set of ablations, with each change good improvements

Weaknesses
- Should have comparisons to other diffusion based methods such as Minimax diffusion or D4M

---

> ### Author Rebuttal · Authors · 2025-07-28
>
> Dear Reviewer,
>
> Thank you for your very positive and encouraging review. We are thrilled that you found the paper to have excellent presentation and clarity, and that you appreciated the directness of our method and the strong empirical results. Your high rating and confidence in our work are greatly appreciated.
> We are happy to answer your questions and address the points you raised to further improve the paper.
>
> > **Are there baseline methods or papers which use guided diffusion rather than directly optimizing z_T?**
>
> Our main comparison is with GLaD because it is the state-of-the-art for our specific paradigm: end-to-end, gradient-based distillation using a generative prior. That said, we do include a direct comparison with D4M on CIFAR-10 (Table 2b), where LD3M shows superior performance, and D4M could be seen as guided diffusion rather than directly optimizing Z.
>
> In general, it would be interesting to see guided diffusion to directly infer a trainable dataset, but our preliminary experiments suggest that this line of research needs more work to be usable in that way.
> Below, we provide our findings for ImageNet-1K (IPC roughly 1200) with ResNet-18 as a classifier (90 Epochs), and it can be seen that, on average, the performance is worse compared to any distillation approach. Interestingly, with more modern LDMs, the performance gets even worse (probably due to recent fine-tuning approaches towards better aesthetics).
>
> |Variant|Final val. acc. (%)|
> |-|:-:|
> |Baseline (real data)	| 70.0 |
> | SD 1.0 | 17.1 |
> | SD 2.1 | 14.5 |
> | SDXL | 9.7|
>
> > **How does the randomness of the denoising process affect LD3M? From Algorithm 1, it seems that because of the sampling of $\epsilon$, you will get a different image every time? Are these samples fixed for each example (making the process deterministic)?**
>
> You are correct that the sampling of $\epsilon$ introduces stochasticity. We fix, however, for one sampling phase, the residual variable $z_T$ to be constantly the same. Thus, we stochastically sample $z_T$ in the first step but keep it constant for all remaining time steps. Therefore, it is not a deterministic process due to its stochastic sampling in the first step.
>
> > **What is the impact of the choice of residual connection in equation (6)? Have the authors experimented with different types of flows?**
>
> This is an important design choice. We found that the simple, parameter-free linear decay was highly effective empirically and robust across different datasets and settings. We chose it for its simplicity and to avoid introducing additional learnable parameters. As we note in our future work section, exploring alternative residual formulations (e.g., exponential or learned schedules) is a promising avenue for future research, which we found orthogonal to the current state and goal of the paper.
>
> > **Does the existing backwards diffusion process not already have some aspect of the skip connection? ...**
>
> This is a very sharp observation. You are right that the standard update rule (Equation 3) contains $z_t$ on the right-hand side, which acts as a "local" skip connection from step $t$ to step $t-1$. However, the vanishing gradient problem arises from the cumulative product of Jacobians across the entire chain of T steps. Our proposed modification is a "global" skip connection that creates a direct path from the final generated latent $z_0$ all the way back to the initial learnable codes, bypassing the long chain. This is what fundamentally preserves the gradient signal and is empirically shown in Table 1 of the paper.
>
> > **Why does it make sense to use the autoencoder as initialization for $z_T$? Isn't the autoencoder supposed to produce values for $z_0$?**
>
> It is both true since the diffusion model operates in the latent space of the autoencoder, so between the encoder (which maps the pixel space to the latent space, i.e., $z_T$) and the decoder (which maps $z_0$ back to pixel space). Thus, in order to have a good initialization of the latent codes, it makes sense to use the encoder to derive the respective latent representations prior to any distillation.

---

> > ### Comment · Reviewer_2fpv · 2025-08-05
> >
> > Thank you for the response to the rebuttal. I hope that these clarifications will make its way into the paper/appendix in future revisions. I will be retaining my score and recommending acceptance.

---

> > > ### Author Response · Authors · 2025-08-08
> > >
> > > We’re pleased we addressed your concerns and grateful for your support in retaining the positive score.

---

### Official Review · Reviewer_qEHp · 2025-07-10

**Clarity:** 3
**Significance:** 2
**Originality:** 2
**Rating:** 4
**Confidence:** 4

**Summary:**

This paper proposes LD3M (Latent Dataset Distillation with Diffusion Models) for end-to-end dataset distillation to prevent vanishing gradients. The authors introduce linearly decaying residual connections from the initial noisy latent state into each reverse diffusion step without requiring diffusion model fine-tuning. Evaluated across multiple ImageNet subsets and CIFAR-10, LD3M outperforms GAN-based methods (GLaD), while demonstrating faster distillation times and simpler initialization compared to GLaD.

**Questions:**

(1) As mentioned above, it is unknown how LD3M scale to distillation of full ImageNet-1K at 224 × 224 resolution and IPC = {10, 50, 100}. These experiments have been done in SRe$^2$L [1], DWA [2], D$^4$M [3], and RDED [4]. Could LD3M still outperform these SOTA methods at the lower cost?

(2) Why is linear decay (in Eq. 6) used instead of exponential or learned schedules? Is this choice empirically or theoretically motivated?

(3) The base model is sometimes from ‘Random’, which is confusing for its definition. Moreover, if the base network is Stable Diffusion trained on LAION (not trained for one specific dataset, such as ImageNet and FFHQ), how does LD3M perform, compared with the current network?

(4) Although the paper makes some claims on privacy and robustness applications, there are no experiments and theoretical analysis to contribute to such statements. Therefore, beyond accuracy, how do LD3M’s distilled samples compare to GLaD in metrics like adversarial robustness or privacy leakage?

(5) The paper optimizes class embeddings but does not explore text prompts as conditioning. Could text-guided distillation improve cross-class discriminability?

(6) Is it possible that an efficient sampling algorithm or few-step model (e.g., DPM-solver and Consistency Models) can achieve better sample-quality in fewer steps?

**Reference:**

[1] Yin, Z., Xing, E., and Shen, Z. Squeeze, Recover and Relabel: Dataset Condensation at ImageNet Scale From A New Perspective. In NeurIPS, 2023.

[2] Du, J., Zhang, X., Hu, J., Huang, W., and Zhou, J. T. Diversity-driven synthesis: Enhancing dataset distillation through directed weight adjustment. In NeurIPS, 2024.

[3] Su, D., Hou, J., Gao, W., Tian, Y., and Tang, B. D$^4$M: Dataset Distillation via Disentangled Diffusion Model. In CVPR, 2024.

[4] Sun, P., Shi, B., Yu, D., and Lin, T. On the diversity and realism of distilled dataset: An efficient dataset distillation paradigm. In CVPR, 2024.

**Ethical Concerns:**

["NO or VERY MINOR ethics concerns only"]

**Final Justification:**

The authors have already addressed my concerns. Thus, I consider increasing my final score to 'Borderline accept'.

**Limitations:**

Yes, the authors adequately discuss technical limitations, including dependence on LDM’s autoencoder structure and trade-offs between diffusion steps and performance.

**Paper Formatting Concerns:**

No.

**Quality:**

2

**Strengths And Weaknesses:**

Strengths:

(1) The proposed method seems technically sound with various experiments across ImageNet subsets, multiple resolutions and IPC settings. Ablation studies convincingly isolate contributions of key components.

(2) The paper is overall easy to follow and well-structured with clear problem formulation and gradient decay analysis. It solves a critical bottleneck (vanishing gradients) to prevent diffusion models from being used in dataset distillation.

Weaknesses:

(1) Experiments are limited to ImageNet subsets and CIFAR-10. It is necessary to show validations on larger-scale datasets (e.g., full ImageNet-1K).

(2) Some of stated advantages remain unsupported by experiments. For example, it lacks evaluations to show maintained or even improved robustness against adversarial attacks or membership inference, which could differ LD3M from other baselines.

---

> ### Author Rebuttal · Authors · 2025-07-27
>
> Dear Reviewer,
>
> We sincerely thank you for your detailed and insightful review. We are grateful that you recognized the technical soundness of our proposed LD3M, the clarity of the paper, and that our work solves the critical bottleneck of vanishing gradients for dataset distillation with diffusion models.
>
> We appreciate the constructive weaknesses and questions you raised. We believe they can be fully addressed. We provide point-by-point responses below.
>
> > **(1) As mentioned above, it is unknown how LD3M scale to distillation of full ImageNet-1K at 224 × 224 resolution and IPC = {10, 50, 100}. These experiments have been done in SRe$^2$L [1], DWA [2], D$^4$M [3], and RDED [4]. Could LD3M still outperform these SOTA methods at the lower cost?**
>
> We agree that evaluation on the full ImageNet-1K dataset is an important long-term goal for any distillation method. However, we respectfully argue that our paper's primary contribution is foundational and, therefore, compared with GLaD, the state-of-the-art in generative prior-based distillation. The methods you cited (SRe2L, DWA, RDED) represent a different, parallel line of research that often relies on statistics matching (to be precise: BN alignment as prior) or other decoupled strategies  and does not use a generative prior for end-to-end optimization. While their large-scale results are impressive (and fast), our work opens a fundamentally new pathway for distillation.
>
> As you correctly noted, scaling is a crucial next step. We explicitly acknowledge this in our paper, stating that evaluation on larger benchmarks like the full ImageNet-1K remains an important direction for future work. Since our work does not rely on dataset statistics (like in SRe2L with BN alignment), we expect that our results (final performance) translate to other classes in ImageNet as every class is distilled in isolation of every other class (great for multi-gpu distillation). By exploiting knowledge distillation and soft-labeling [5], we would expect even greater performance similar to the aforementioned methods. However, we want to note that the cited works are explicitly designed to be fast, which we believe is hard to beat with respect to distillation speed (e.g., RDED is essentially a concatenation of real-images, which is incredible fast in comparison).
>
> Yet, despite their fastness, they are still comparable with more straightforward methods like MTT/TESLA. See the RDED paper as example: MTT/TESLA reaches 7.7±0.2 and 17.8±1.3, whereas SRe2L reaches 0.1±0.1 and 21.3±0.6 on IPC=1 and 10, respectively on ImageNet-1K. MTT/TESLA can be used on top of our generative prior and improved upon with that strategy.
>
> [5] Tian Qin, Zhiwei Deng, David Alvarez-Melis: A Label is Worth a Thousand Images in Dataset Distillation, In NeurIPS, 2024.
>
> > **(2) Why is linear decay (in Eq. 6) used instead of exponential or learned schedules? Is this choice empirically or theoretically motivated?**
>
> A linear schedule provides a simple, parameter-free, and interpretable way to bridge the gradient flow. It ensures that the influence of the initial noisy state is strongest at the beginning of the reverse process (when the denoiser's prediction is less reliable) and smoothly fades as the denoised, final state is formed. Thus, it was chosen (theoretically motivated) for its simplicity. We agree that other schedules could be explored. We explicitly mention in our "Future Work" section that investigating "alternative residual formulations"  is a promising direction.
>
> > **(3) The base model is sometimes from ‘Random’, which is confusing for its definition. Moreover, if the base network is Stable Diffusion trained on LAION (not trained for one specific dataset, such as ImageNet and FFHQ), how does LD3M perform, compared with the current network?**
>
> We apologize for the confusion. "Random" in Table 5 refers to a generator (in this case, StyleGAN-XL for GLaD and the LDM for our method) whose weights are randomly initialized instead of being pre-trained on a specific dataset like ImageNet or FFHQ. This is a standard ablation to test the importance of the pre-trained prior. Our results in Table 5 show that LD3M is robust and performs well even with a random prior. So, we expect only marginal improvements with a DM trained on LAION. Nevertheless, using a general text-to-image model like a newer Stable Diffusion model would be a very exciting future direction, which leads directly to your next question.
>
> > **(5) The paper optimizes class embeddings but does not explore text prompts as conditioning. Could text-guided distillation improve cross-class discriminability?**
>
> This is a great point. The extension of LD3M to use text prompts is a natural and powerful next step. We focused on learnable class embeddings in this work to maintain a controlled comparison against GLaD. This allowed us to isolate the specific benefits of using a diffusion prior over a GAN prior. However, our framework is indeed general. Replacing the learnable class embeddings with text embeddings from a model like CLIP is a straightforward extension that could enable distillation of datasets with more rich textual descriptions, potentially improving cross-class discriminability as you suggest. We believe this is a key avenue for future research spawned by our work. A great starting point could be the following work: [6]
>
> [6] Stanislav Fort, Jonathan Whitaker: Direct Ascent Synthesis: Revealing Hidden Generative Capabilities in Discriminative Models, In arXiv, 2025.
>
> > **(4) Although the paper makes some claims on privacy and robustness applications, there are no experiments and theoretical analysis to contribute to such statements. Therefore, beyond accuracy, how do LD3M’s distilled samples compare to GLaD in metrics like adversarial robustness or privacy leakage?**
>
> We acknowledge that the phrasing in the introduction could be clearer. Our intention was not to claim that LD3M achieves superior robustness or privacy, but rather that it enables the gradient-based optimization necessary to even explore these properties.
> The core motivation for pursuing gradient-matching optimization is its potential for fine-grained control, which is crucial for applications like enhancing adversarial robustness or privacy. For instance, see [7] as an example, where a modified MTT for improved robustness is proposed, which can be applied on top of our generative prior. Previous diffusion-based methods like D4M, which rely on sampling, cannot perform these types of optimization. LD3M is the first diffusion-based method that would make this possible.
>
> The scope of our paper is to solve the foundational vanishing gradient problem. A full evaluation of all potential downstream benefits (like adversarial robustness or privacy leakage) is substantial enough for follow-up work and beyond the scope of this initial paper.
>
> [7] Lai et al., Robust Dataset Distillation by Matching Adversarial Trajectories. In arXiv, 2025.
>
> > **(6) Is it possible that an efficient sampling algorithm or few-step model (e.g., DPM-solver and Consistency Models) can achieve better sample-quality in fewer steps?**
>
> This is another excellent and forward-looking question. Our work strongly suggests that fewer steps are indeed better for the distillation task.
> Our analysis on the number of diffusion steps T (Figure 5) clearly shows that LD3M's performance peaks around T=10-40 and does not improve (and can even slightly decrease) with more steps. This indicates that a long, costly diffusion chain is not required.
>
> We explicitly identify this as a key area for future work. In our conclusion, we state that future work should explore "integration with fast samplers like DPM-Solver". Such an integration would further improve LD3M's efficiency, reducing the distillation time and memory usage reported in our paper and, ultimately, enhancing its practical appeal.

---

> > ### Comment · Reviewer_qEHp · 2025-08-08
> >
> > I really appreciate authors‘ feedback to all my questions. I am satisfied with most of the answers except for Q(1). In fact, there is a recent work [1] that utilizes Diffusion Model for Dataset Distillation and achieves SOTA results. Could authors further compare LD3M with this work [1] and analyze the connections or differences from the perspective of methodology?
> >
> >
> > [1] Taming Diffusion for Dataset Distillation with High Representativeness, Zhao et al., ICML 2025.

---

> > > ### Author Response · Authors · 2025-08-08
> > >
> > > Dear Reviewer,
> > >
> > > thank you for the follow-up question. It is true that both work share the commonality of exploiting a diffusion prior for dataset distillation. However, there are significant differences in the working mechanisms, which gives different guarantees for other motivations of using dataset distillation (such as adversarial robustness or privacy, which are only available for optimization-based distillation methods). In more detail, we want to summarize both approaches below:
> > >
> > > **LD3M (ours): Optimization-based**. It enables end-to-end gradient learning of distilled latents and class embeddings through a frozen latent diffusion model by injecting a linearly decaying skip from the initial noise into every reverse step, which fixes vanishing gradients without finetuning the denoiser.
> > >
> > > **D3HR: Sampling/selection-based**. It DDIM-inverts VAE latents to a high-normality noise domain, fits a per-class Gaussian, then group-samples many candidate subsets and selects the one whose statistics (mean/std/skew) best match the target; finally DDIM-samples back to images. This addresses distribution mismatch, randomness from initial noise, and per-sample “separate” sampling.
> > >
> > > What is learned:
> > > - LD3M: Latent codes and condition embeddings are learned (optimized).
> > > - D3HR: Learns nothing inside the diffusion model; it estimates per-class Gaussian stats ( $\mu$, $\sigma$; skew for scoring) and selects the best subset via a metric.
> > >
> > > In summary, both approaches are complementary: LD3M emphasizes learned synthetic data via gradient-flow engineering, whereas D3HR emphasizes distributional representativeness via statistical modeling and selection in diffusion’s noise domain.

---

> > > > ### Comment · Reviewer_qEHp · 2025-08-08
> > > >
> > > > Thank the authors for their responses. Please incorporate all these discussions into future revisions of the paper.

---

### Decision · Program_Chairs · 2025-09-17

**Decision:**

Accept (spotlight)

**Comment:**

(a) The paper introduces LD3M, an end-to-end dataset distillation method in the latent space of a pretrained LDM to prevent vanishing gradients. The key idea is to add linearly decaying residual (skip) connections from the initial noisy latent z_T into each reverse diffusion step. LD3M can be used with existing distillation objectives, outperforms GLaD on several datasets, and is efficient.

(b) The method is simple and relatively general (adding skip connections), it addresses the problem of vanishing gradients and enables the use of diffusion models for database distillation, it performs better than GLaD on several datasets. The reviewers also found the paper to be well-written.

(c) The reviewers raised a number of concerns: 1) The evaluation scope is found limited and thus the scalability is unclear; 2) No experiments to support robustness claims; 3) Some technical details are not clearly described in the paper; 4) Missing evaluation of the computational load.

(d) All reviewers agree that this paper should be accepted and find the contribution solid. The proposed method is indeed simple but effective and would be of interest to the general conference audience. It is not recommended as an oral because the paper needs some more work to integrate more extensive comparisons and robustness evaluations.

(e) The reviewers raised several concerns but these were mostly addressed by the authors and the reviewers found them satisfactory. The concerns of qEHp, 2fpv, 2brr were on missing experiments on the full ImageNet-1K and recent diffusion-based distillation comparisons. There were several concerns on the technical details of the method by all reviewers (eg, linear decay, relation to existing methods,  stochasticity/determinism, and optimizer and initialization choices). There were concerns on the presentation (eg, equations and tables).
The authors addressed all these concerns. Moreover, based on the fact that reviewers had no follow up concerns and were satisfied with the answers, I decided that the paper should be accepted (and the authors should incorporate their answers into the final version of the paper).